# Snow albedo sensitivity to macroscopic surface roughness using a new ray tracing model

Fanny Larue[1], Ghislain Picard[1], Laurent Arnaud[1], Inès Ollivier[1], Clément Delcourt[1], Maxim Lamare[1,2], François Tuzet[1,2], Jesus Revuelto[2], Marie Dumont[2]

[1]UGA/CNRS, Institut des Géosciences et de l'Environment (IGE), Grenoble, 38100, France

[2]Univ. Grenoble Alpes, Université de Toulouse, Météo-France, CNRS, CNRM, Centre d'Études de la Neige, Grenoble, France

*Correspondence to*: Fanny Larue (fanny.larue@univ-grenoble-alpes.fr)

**Abstract.** Most models simulating snow albedo assume a flat and smooth surface, neglecting surface roughness. However, the presence of macroscopic roughness leads to a systematic decrease in albedo due to two effects: 1) photons are trapped in concavities (multiple reflection effect) and, 2) when the sun is low, the roughness sides facing the sun experience an overall decrease in the local incident angle relative to a smooth surface, promoting higher absorption, whilst the other sides has weak contributions because of the increased incident angle or because they are shadowed (called the effective angle effect here). This paper aims to quantify the impact of surface roughness on albedo and to assess the respective role of these two effects, with 1) observations over varying amounts of surface roughness, and 2) simulations using the new Rough Surface Ray Tracer (RSRT) model, based on a Monte Carlo method for photon transport calculation.

The observations include spectral albedo (400-1050 nm) over manually-created roughness surfaces with multiple geometrical characteristics. Measurements highlight that even a low fraction of surface roughness features (7 % of the surface) causes an albedo decrease of 0.02 at 1000 nm when the solar zenith angle ($\Theta_s$) is larger than 50°. For higher fractions (13 %, 27 % and 63 %), and when the roughness orientation is perpendicular to the sun, the decrease is of 0.03 – 0.04 at 700 nm and of 0.06 – 0.10 at 1000 nm. The impact is 20% lower when roughness orientation is parallel to the sun. The observations are subsequently compared to RSRT simulations. Accounting for surface roughness improves the model observation agreement by a factor two at 700 nm and 1000 nm (errors of 0.03 and 0.04, respectively), compared to simulations considering a flat smooth surface. The model is used to explore the albedo sensitivity to surface roughness with varying snow properties and illumination conditions. Both multiple reflections and the effective angle effect have more impact with low SSA (< 10 m$^2$ kg$^{-1}$). The effective angle effect also increases rapidly with $\Theta_s$ at large $\Theta_s$. This latter effect is larger when the overall slope of the surface is facing away the sun and with a roughness orientation perpendicular to the sun.

For a snowpack where artificial surface roughness features were created, we showed that a broadband albedo decrease of 0.05 may cause an increase of the net short wave radiation of 80 % (from 15 W m$^{-2}$ to 27 W m$^{-2}$). This paper highlights the necessity to consider surface roughness in the estimation of the surface energy budget and opens the way to consider natural rough surfaces in snow modeling.

## 1 Introduction

Spectral albedo quantifies the proportion of solar energy reflected by a surface for each wavelength, and governs the quantity of solar radiation absorbed in the snowpack. Because snow has an overall high albedo in the solar spectrum, a small decrease in albedo (e.g. from 0.85 to 0.75) drastically increases the proportion of absorbed energy (from 25% to 15%; Genthon, 1994). Thus, a reduction in albedo has important consequences on the surface energy budget, impacting surface temperature (Mondet and Fily, 1999, Picard et al. 2012, Fréville et al., 2014), and the hydrology of watersheds (e.g. Flanner et al., 2009; Painter et al., 2010; Oaida et al., 2015). Several studies have investigated the spatial and temporal variability of snow albedo using in-situ data (Brock et al., 2000; Wuttke et al., 2006; Dumont et al., 2017) or satellite observations (Atlaskina et al., 2015; Naegeli

& Huss, 2017). Snow spectral albedo generally depends in a complex way on several factors, including 1) the snow physical and chemical properties, mainly the Specific Surface Area of snow grains (SSA, Gallet et al., 2009), the snow grain shapes (Tanikawa et al., 2006; Jin et al., 2008; Libois et al., 2013, 2014) and the concentration of snow Light Absorbing Particles (referred to as LAP, Skiles et al., 2018), 2) the spectral and angular characteristics of the incident radiation (Warren, 1982), 3) the presence of macroscopic surface roughness (Kuhn, 1985; Warren et al., 1998; Mondet and Fily, 1999). The first two points

have been thoroughly studied, showing that for a smooth surface, snow albedo decreases as SSA lowers (coarsening snow granularity) and with a higher sun elevation (i.e. a decrease in solar zenith angle), both of which lead to an increased absorption (Warren et al., 1998, Kokhanovsky and Zege, 2004). Nevertheless, the effects of roughness are often neglected due to the difficulty to characterise the actual surface roughness within the footprint of the sensor.

Snow-covered surfaces often exhibit macroscopic roughness, resulting from snow transport or erosion by the wind or snow

melting (Filhol and Sturm, 2015). In Antarctica, roughness height ranges from a few centimetres to a few meters (Warren et al., 1998; Wuttke et al., 2006), and the features' axis is usually aligned along the prevailing wind direction (Furukawa et al., 1996), whereas in alpine areas the spatial distribution of macroscopic roughness mainly depends on topography, which drives wind direction and its intensity (Naaim-Bouvet et al., 2011). Kuhn (1974) was the first to report a reduction of the forward peak of the Bidirectional Reflectance Distribution Function (BRDF) over a sastrugi field, and attributed this fact to shadows

when the solar azimuth angle is perpendicular to the sastrugi. This motivated further studies that showed a systematic albedo decrease in presence of roughness (Caroll and Fitch, 1981; Leroux and Fily, 1998; Corbett and Su, 2015). The amplitude of the reduction in albedo depends on illumination conditions, snow properties, the size and the orientation of roughness features (Hudson and Warren, 2007; L'Hermitte et al., 2014). For instance, in high altitude mountain glaciers, the presence of penitentes, which can reach several meters in height (Lliboutry, 1953), causes a measured albedo decrease of 8-10% (Corripio and Purves,

2006). These studies underlined the difficulty of precisely quantifying the impact of roughness since the illumination conditions and snow properties also vary during albedo measurements, making it difficult to evaluate the reduction in albedo due to roughness only. A protocol was proposed by Kuchiki et al. (2011) using a controlled environment where the precise roughness shapes, orientation and dimensions, snow properties and illumination conditions were known. Over a manually-created artificial roughness field, they showed that the hemispherical-directional reflectance (HRDF) factor varies by more than ± 50%

relative to a smooth surface. Nevertheless, they did not acquire albedo measurements, i.e. bi-hemispherical reflectance.

Warren et al. (1998) showed that the albedo decrease over a roughness field is controlled by two effects: 1) a decrease in the insolation-weighted average incidence angle relative to a flat surface (further referred to as the effective angle effect), and 2) multiple reflections in the concavities. The first effect is explained by the fact that the sides of the roughness shapes facing the sun experience stronger radiation with a smaller angle than the solar zenith angle which enhances absorption in the case of

snow surface (Warren, 1982), and the sides facing away from the sun receive less radiation due to shadows or grazing angles. The insolation-weighted average albedo is therefore reduced relatively to a flat and smooth surface (Warren, 1982; 1998; Kokhanovsky and Zege, 2004). The effective angle effect varies with the shape, size, and orientation of the roughness features (Carroll and Fitch, 1981; L'Hermitte et al., 2014), and is significant under direct illumination and for low sun elevations only (Warren et al., 1998). The second effect of roughness involves multiple reflections cause by the trapping of photons between

roughness shapes (Pfeffer and Bretherton, 1987). Over a smooth surface, a photon only hits the surface once and is either absorbed or reflected to the sky. Over a rough surface, photons can not only be absorbed or reflected to the sky, but they can also be reflected back to the surface. In this latter case, they have another probability to be absorbed, at every hit. This results in a systematic increase in absorption, and thus a decrease in albedo. The impact is maximal when the probabilities of reflection and absorption are balanced, i.e. for intermediate values of albedo (close to 0.5 in the near infrared at 700-1100 nm). Instead

in the visible where albedo is close to 1, the probability of absorption is too low to trap the photons, and oppositely in the mid-infrared where the albedo is close to 0, the impact of multiple reflections is negligible. This trapping effect operates under

direct and diffuse illumination. Although these two effects have never been quantified separately, Warren et al. (1998) suggested to acquire measurements in diffuse illumination to estimate the impact of multiple reflections only.

Photometric models based on analytical equations were developed to simulate the effects of roughness on albedo using idealized geometric shapes (Carroll, 1982; Pfeffer and Bretherton, 1987; Wendler and Kelley, 1988; Leroux and Fily, 1998; Cathles et al., 2011; 2014; Zhuravleva and Kokhanovsky; 2010, 2011). Leroux and Fily (1998) predicted a decrease in albedo over a sastrugi field of 5 - 9 % at 900 nm, depending on the sastrugi orientation with respect to the sun position. Despite their interest to draw general conclusions on the albedo sensitivity to roughness characteristics, these models are of limited interest for real roughness features due to the idealization of the shapes (Warren et al., 1998). In addition, they use the Lambertian approximation to represent the surface reflectivity, and do not consider the intrinsic BRDF of the snow, meaning that they cannot simulate the effective angle effect. To explore the real impact of surface roughness, a 3D radiative transfer model is needed. Monte Carlo photon transport algorithms are convenient approaches (Lafortune, 1995; O'Rawe, 1991; Iwabuchi, 2006; Kuchiki et al., 2011). However, most studies using these numerical methods aim to evaluate the BRDF or HRDF instead of albedo, as their application domain was remote sensing (Kuchiki et al., 2011; L'Hermitte et al., 2014; Corbet et al., 2015).

The aims of this paper are two-fold: 1) to quantify the impact of surface roughness on snow albedo, as a function of roughness features, illumination conditions and snow properties, and 2) to assess the respective roles of the effective angle effect and multiple reflections with a new model able to represent surface roughness. Firstly, we collected albedo measurements in controlled experiments following the idea of Kuchiki et al. (2011). We produced various artificial rough surfaces during four field campaigns in the French Alps in 2018 and 2019 (Sect. 2). In each experiment, albedo measurements were acquired for several illumination conditions and with numerous geometrical characteristics at the surface. Observations were also acquired over nearby smooth surfaces to serve as references. Secondly, we developed a new model based on the Monte Carlo photon transport method, the Rough Surface Ray Tracing model (RSRT), to simulate albedo by considering surface roughness (Sect. 3). RSRT was evaluated using the albedo observations (Sect. 4.1). In Section 4.2, the model was used to explore the albedo sensitivity to surface roughness according to SSA, terrain slope, roughness orientation and solar zenith angle. The model was applied to assess the respective roles played by the effective angle effect and multiple reflections (Sect. 4.3). At last, the sensitivity of the net short wave radiation to the presence of surface roughness is discussed to estimate the potential impact on the surface energy balance (Sect. 4.4).

## 2 Field experiments

*In situ* measurements of albedo were acquired in the French Alps over smooth and rough snow surfaces. This section details how the rough surfaces were created, and measurements acquired in the field.

### 2.1 Artificial rough snow surfaces

Artificial rough snow surfaces were created by delineating squares of 2.5 x 2.5 m$^2$. Roughness features were manually created on natural smooth surfaces, by varying their number and orientation. The features were produced parallel to each other, regularly spaced with a period $\Lambda$, and with an azimuth angle $\varphi_r$, taken clockwise from the North. The roughness orientation with respect to the solar azimuth angle ($\varphi_s$) was defined by $\Delta\varphi_r$, the difference $\varphi_s - \varphi_r$. Figure 1 shows the experimental setup and the variables involved. Each surface was characterised by its aspect, its slope, and its roughness properties (number, shape, size and orientation). Two types of experiments were performed:

    a) Sensitivity to the fraction of roughness features:

The fraction of roughness features in the 2.5 x 2.5 m$^2$ area is described with the width-to-period ratio $\eta$ *(i.e. $\eta = W/\Lambda$, expressed in percentage, where $W$ is the width of roughness shapes). The albedo sensitivity to $\eta$ was studied during two experiments at

the Col du Lautaret site (45°2' N, 6°2' E, 2100 m a.s.l.) over two different dates (06 April 2018 and 17 April 2018), respectively called experiments A and B. Figure 2 illustrates the field experiment A, and Table 1 details the characteristics, acronyms and parameters for each studied surface. Snow albedo was first measured over the smooth surface (called A-smooth and B-smooth-dry in Table 1), and then the roughness shapes were created in the smooth surface by uniformly pressing a rectangular metal

bar into the snow ($H$ = 2 cm depth and $W$ = 4 cm width), in the North-South direction ($\varphi_r$ = 0°-180°). The rectangular shapes were created with a period $\Lambda$ = 55 cm (5 shapes over 2.5 m, $\eta$ = 7 %). After albedo measurements were acquired, identical rectangular shapes were added to reach a period $\Lambda$ = 30 cm (10 shapes over 2.5 m, $\eta$ = 13 %), then $\Lambda$ = 15 cm (20 shapes over 2.5 m, $\eta$ = 27 %) (Figure 2 and Table 1). Because it takes approximately one hour to make a series of measurements, the increasing fraction of roughness features is correlated to solar zenith angle ($\Theta_s$) variations that also change albedo. To attempt

to decouple the two effects, experiment A was conducted when the sun was going down ($\Theta_s$ went from 56.6° to 63.7°), whereas in the experiment B, the sun was going up ($\Theta_s$ went from 56.0° to 40.0°).

Other changes may also occur during that time. In Experiment B for instance, melting was observed on the B-$\eta27\%$ surface (the sun was close to the nadir), which leads to an increase in snow wetness and a decrease in surface SSA compared to the B-smooth-dry surface analysed at the beginning of the day. To allow more reliable comparisons, we simultaneously measured

albedo over the B-$\eta27\%$ surface, and a nearby smooth surface (called B-smooth-wet in Table 1).

  b) Sensitivity to the roughness orientation:

The albedo sensitivity to roughness orientation was studied with two experiments at the Arcelle site (45°6' N, 5°52' E, 1729 m a.s.l.) over two dates (11 January 2019 and 22 February 2019), respectively called experiments C and D. The roughness shapes were triangular, $H$ = 6 cm depth and $W$ = 7 cm width, and created with a period $\Lambda$ = 11 cm ($\eta$ = 63 %). Fig. 1b and Table 1

detail the experimental setup.

In Experiment C, measurements were simultaneously acquired every 20 minutes over a surface with roughness features oriented at $\varphi_r$ = 90° (called C rough 90°), another one with roughness features at $\varphi_r$ = 0° (called C rough 0°), and a smooth surface for reference (called C smooth). In Experiment D, only two surfaces were compared every 20 minutes: a rough surface with roughness features at $\varphi_r$ = 90° (called D rough 90°), and a smooth surface (called D-smooth). For both experiments,

studied surfaces were close enough to consider that snow properties evolved with the same dynamics. Note that it took about up to 5 minutes to acquire one set of albedo measurements, and to move to the next surface. Measurements were acquired all day in the experiment C (sun going up and down), and during the morning in Experiment D (sun going up only).

  The albedo sensitivity to roughness features is quantified by comparing rough and smooth surfaces for each experiment.

  In the reality, it is difficult to find perfectly flat surfaces, and all studied surfaces have small slopes. In particular, it is

noteworthly that experiments A, B and C have a small sun-facing slope.

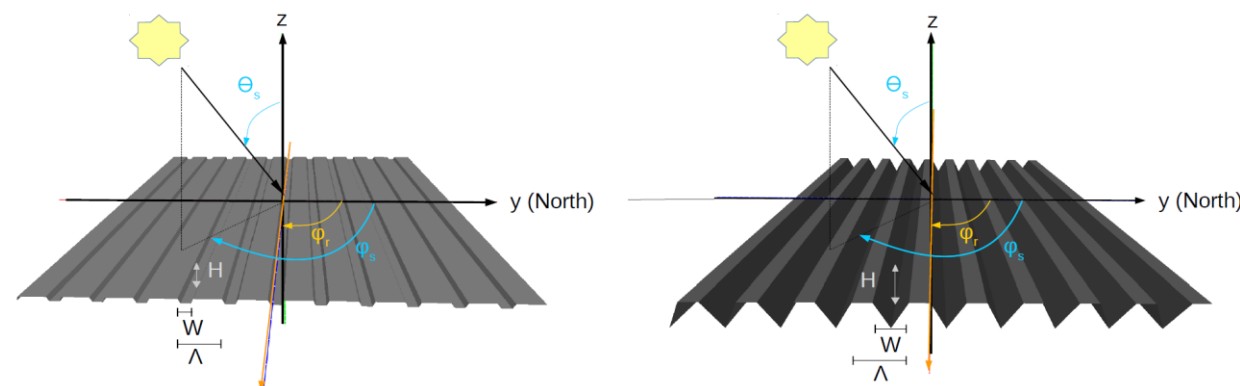

**Figure 1. Illustration of the setups for a) A-η27% and B-η27% experiments, and b) C rough 90° and D rough 90° experiments (Table 1 for acronyms). The grey surfaces are modelled meshes with parallel shapes (rectangular or triangular), similar to the artificial**
**roughness surfaces created in the field. The two sites are areas of 2.5 x 2.5 m². H is the height, W the width, and $\Lambda$ the period of roughness features. $\varphi_r$ is the roughness orientation, $\varphi_s$ the solar azimuth angle, and $\theta_s$ the solar zenith angle. Azimuth angles are clockwise from North (y axis).**

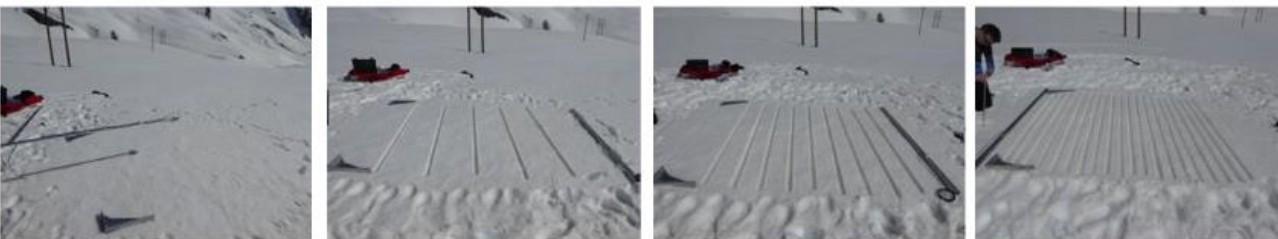

**Figure 2. Studied surfaces of experiment A (Table 1), from left to right: A-smooth, A-η7%, A-η13%, and A-η27% sites.**

**Table 1. Details of field experiments. The sensor's height is fixed at 65 cm.**

| Location | #field experiment | Acronyms | SSA [m²/kg] | $\Delta\varphi_r$ [°] | $\eta$ [%] | $\varphi_s$ [°] | $\theta_s$ [°] | Slope $\theta_n$ [°] | Aspect $\varphi_n$ [°] | Characteristics |
|---|---|---|---|---|---|---|---|---|---|---|
| **Col du Lautaret (April 2018)** | **A** | A-smooth | 7.4 | - | 0 | 242 | 56.6 | 3.1 | 216 | Naturally flat smooth surface |
| | | A-η7% | 7.4 | 64 | 7.0 | 244 | 57.5 | 3.1 | 216 | 5 Rectangular shapes |
| | | A-η13% | 7.4 | 69 | 13 | 249 | 61.8 | 3.1 | 216 | 10 Rectangular shapes |
| | | A-η27% | 7.4 | 71 | 27 | 251 | 63.4 | 3.1 | 216 | 20 Rectangular shapes |
| | **B** | B-smooth-dry | 4.5 | - | 0 | 112 | 55.5 | 3.6 | 105 | Naturally flat smooth surface. Dry snow conditions |
| | | B-η7% | 4.5 | 73 | 7 | 118 | 51.2 | 3.6 | 105 | 5 Rectangular shapes |
| | | B-η13% | 4.5 | 63 | 13 | 129 | 45.5 | 3.6 | 105 | 10 Rectangular shapes |
| | | B-η27% | 4.5 | 49 | 27 | 142 | 40.0 | 3.6 | 105 | 20 Rectangular shapes |
| | | B-smooth-wet | 4.5 | - | - | 159 | 36.4 | 3.2 | 96.0 | Naturally flat and smooth surface. Wet snow conditions |
| **Arcelle (January-February 2019)** | **C** | C smooth | 86.0 – 100 | - | 0 | 137 – 211 | 66.1 – 78.9 | 3.3 | 169 | Naturally flat smooth surface. |
| | | C rough 90° | 86.0 – 100 | 48.5 – 121 | 63 | 139 –211 | 66.9 – 78.1 | 3.3 | 166 | 20 Triangular shapes oriented at $\varphi_r = 90°$ |
| | | C rough 0° | 86.0 – 100 | -21.0 – 21.0 | 63 | 159 – 199 | 67.2 – 69.6 | 4.0 | 150 | 20 Triangular shapes oriented at $\varphi_r = 0°$ |
| | **D** | D smooth | 4.8 - 8.9 | - | 0 | 132-177 | 55.3 - 68.5 | 1.8 | 246 | Naturally flat smooth surface. |
| | | D rough 90° | 4.8 - 8.9 | 41.0 – 73.0 | 63 | 131-161 | 55.3 - 65.1 | 1.4 | 281 | 20 Triangular shapes oriented at $\varphi_r = 90°$ |

## 2.2 Spectral albedo measurements

Spectral albedo, or more precisely the bi-hemispherical reflectance (Schaepman-Strub et al., 2006), is the ratio of the upwelling and the downwelling spectral irradiance. Snow spectral albedo measurements were acquired with the Solalb instrument, a manual version of the albedometer AutoSolexs described by Picard et al. (2016). Solalb is a hand-held instrument
using a single light collector with a near-cosine response and equipped with an inclinometer located at the end of a 3 m boom. The boom was rotated by the operator to successively acquire the downward and upward solar radiation with a horizontal sensor (± 0.1° accuracy). This operation usually takes up to a maximum time of 30 seconds. Variations of incident illumination caused by clouds between two acquisitions were also measured with a photodiode receiving ambient radiation. Only spectra with stable incident illumination within 1 % were selected. Spectra were acquired over the 400-1050 nm wavelength range
with an effective resolution of 3 nm. The height of the sensor impacts the measured roughness effects, by changing the footprint of the sensor (L'Hermitte et al., 2014). To study this sensitivity, albedo was measured with sensor heights of 45 cm, 55 cm and 65 cm, in the experiments A and B (not shown). We found a weak influence on measured albedo (0.4 ± 0.5 % of differences between spectra), showing that this sensitivity was negligible given the type of roughness considered here, and the sensor's height. Therefore, the sensor was set to 65 cm high for all experiments. At this height, the footprint is about 2.3 x 2.3 m² (99
% of the signal is coming from a viewing angle of 60°, Picard et al., 2016). The ratio of diffuse-to-total irradiance ($r_{diff\text{-}tot}$) was also measured shortly after the albedo measurement by screening the sun to record the diffuse irradiance, the total irradiance being measured with the sensor looking upward.

Post-processing was applied to each acquired spectrum following Picard et al. (2016). This includes dark current correction, considering the integration time, and the correction of the collector angular responses.

The observed apparent albedo, hereinafter referred to as $\alpha_{obs}$, is the processed spectrum measured with Solalb, considering the sensor in a horizontal position (Sicart et al., 2001).

The accuracy of $\alpha_{obs}$ mainly depends on that of the levelling of the arm. To estimate $\alpha_{obs}$ uncertainties, measurements were duplicated three times for 6 different sites. A maximal variation of 1.6 % was estimated between the $\alpha_{obs}$ spectra acquired in same field conditions.

## 2.3 Snow surface properties

Snow SSA was measured at the surface using the Alpine Snowpack Specific Surface Area Profiler (ASSSAP) instrument that has an accuracy of 10 % (Arnaud et al., 2011). For the two experiments A and B, we measured the surface SSA in the middle of the experiment (corresponding to $\eta = 13\%$), and the SSA was assumed to be constant throughout the experiments (3 hours). The albedo sensitivity to SSA variations and associated uncertainties is discussed in Sect. 4.2 in order to untwine these contributions from those of roughness. Note that compacting to create the roughness features may have lowered the SSA locally. As the compaction was small (2 cm depth), and as the SSA values were initially low over the studied surfaces, we assumed here that the effect of the compaction on the observed albedo is negligible. For the two experiments C and D, three SSA measurements were taken at the surface at each albedo acquisition: two in the cavities (one over the side facing the sun, one over the side facing away the sun) and one over the smooth surface between cavities. The standard deviations of these three SSA are always lower than 10% of the mean SSA, showing that the compaction effect is negligible compared to measurements uncertainties. The mean of these three SSA values is used in our albedo simulations.

To limit the scope of this study, the concentration of Light Absorbing Particles (called LAP), such as mineral dust and black carbon, was not measured although they strongly lower the spectral signature in the visible range (Warren, 1982), especially at the end of the season when the concentration of impurities is high at the surface (Flanner et al., 2009). It was the case for experiments A and B (measurements acquired in April). Figure 3 shows the spectrum measured over the A-smooth surface. The albedo decrease in the 400-600 nm range is a clear signature of the presence of snow impurities. Even a small amount of LAP led to a high decrease of the albedo in the visible domain (Tuzet et al., 2019). This sensitivity is well described in Dumont et al. (2017). To minimize this contribution, we chose to quantify effects of roughness in the 600-1050 nm wavelength range.

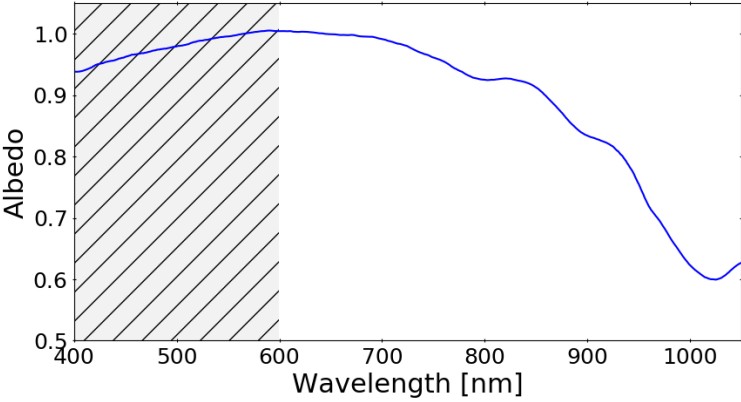

**Figure 3. Measured spectral albedo from 400 nm to 1050 nm. The grey area with vertical lines (from 400 nm to 600 nm) is the wavelength range the most affected by the concentration of snow LAP (impurities). The spectrum is the one acquired over the A-smooth site (Table 1).**

**2.4 Surface slope effects on the measured albedo**

In the case of a tilted surface, Solalb is not perfectly parallel to the snow surface, and therefore the ratio of values acquired by the sensor when it measures the downwelling and the upwelling spectral irradiance (called the apparent or measured albedo, here $\alpha_{obs}$) differs from the intrinsic surface albedo (called true albedo in previous studies, Picard et al., 2020). Indeed, when the sensor is horizontal, the titled surface receives sun radiation with a different incidence angle and is viewed with a reduced solid angle by the sensor (Grenfell et al, 1994; Wutttke et al., 2006; Dumont et al., 2017). With surfaces having a sun-facing slope, it has been demonstrated that measured albedo values may be over 1 in the visible range, because there is a higher interception probability of the sun beam by these slopes facing the sun compared to the horizontal sensor (Picard et al., 2020). Therefore, apparent albedo may exceed 1 in the visible range. In contrast, the intrinsic albedo is strictly ranged between 0 and 1.

In this study, surfaces of experiments A, B and C have small sun-facing slopes (Table 3), and the slope effects can not be neglected in albedo simulations since even a small slope (2°) facing the sun may induce a variation in measured albedo by up to 5 % over a smooth surface (Dumont et al., 2017).

In the field, an inclinometer fixed at the end of a 2 meters ruler was used to measure the slope in the sensor's footprint. The aspect of the slope is defined as the azimuthal direction of the steepest slope, clockwise in degrees from North. However, as the studied surfaces were chosen as flat as possible, the steepest inclination was not visually detectable. Thus, a first inclination measurement was acquired with the ruler parallel to the roughness features, and a second one by rotating the ruler by 90°, in order to estimate the normal of the surface ($\vec{n} = (n_x, n_y, n_z)$). The slope and the aspect were deduced as follow:

$$\theta_n = arccos(n_z) \tag{1},$$

$$\varphi_n = - arctan(n_y/n_x) + \pi/2 \tag{2},$$

where $\theta_n$ is the steepest slope angle, and $\varphi_n$ is the aspect of the slope. In this study, all surface slopes were below 5°. The uncertainty of slope measurements was estimated of $\pm$ 1° due to natural ripples of the studied surfaces. The impact of this uncertainty in our roughness analysis is discussed in Sect. 4.3.2.

**3 A 3-D Monte Carlo radiative transfer Model**

The RSRT model was developed to simulate snow albedo considering macroscopic surface roughness. This combines both 1) the asymptotic radiative transfer theory (Sect. 3.1) to compute the spectral albedo each time a photon hits the modeled surface and 2) a Monte Carlo technique (Sect. 3.2) to estimate the geometric effects introduced by roughness and represented with a 3-D mesh of the studied area. Section 3.3 details the simulation framework and the sensitivity analysis. A simple approach is applied to illustrate the impact of roughness on the quantity of energy absorbed in the snowpack (Sect. 3.4).

**3.1 Asymptotic radiative transfer theory**

In the RSRT algorithm, an ensemble of photons is launched over a modeled surface. This surface is represented with a triangular mesh composed of small facets. Both the spectral albedo and the BRDF distribution are computed for each facet hit by a photon. The Asymptotic Radiative Transfer theory (ART) provides analytical equations to estimate spectral albedo for highly reflective materials, which applies well to snow in the visible and the NIR domains, typically from 400 nm to 1100 nm (Zege et al., 1991; Kokhanovsky and Zege, 2004). Several models use this theory (Negi et al., 2011; Libois et al., 2013; Wang et al., 2017), which is based on three assumptions: 1) the snowpack is represented with vertically and horizontally homogeneous plane-parallel layers, 2) the surface is perfectly smooth and horizontal (flat), 3) single scattering albedo and the snow phase function are described with the asymmetry factor, *g*, the absorption enhancement parameter, *B*, and the SSA of the snow. The albedo simulated with the ART theory have shown a good accuracy compared to observations over smooth surfaces

(Dumont et al., 2017; Wang et al., 2017). The facets of the mesh are small enough to be considered as smooth surfaces. The direct and the diffuse part of the albedo at the wavelength $\lambda$ and $\theta_s$, $\alpha_{dir}(\lambda, \theta_s)$ and $\alpha_{diff}(\lambda)$, are estimated with Eq. (3) and (4):

$$\alpha_{diff}(\lambda) = exp\left(-4\sqrt{\frac{2B\gamma(\lambda)}{3\rho_{ice}SSA(1-g)}}\right) \hspace{2cm} [3],$$

$$\alpha_{dir}(\lambda, \theta_s) = exp\left(\frac{-12(1+2cos\theta_s)}{7}\sqrt{\frac{2B\gamma(\lambda)}{3\rho_{ice}SSA(1-g)}}\right) \hspace{1cm} [4],$$

where $\rho_{ice} = 917$ kg m$^{-3}$ is the bulk density of ice at 0°C, $\gamma(\lambda)$ is the wavelength-dependent absorption coefficient of ice, taken from Picard et al. (2016) here. B and g are the snow shape coefficients and are assumed to be constant. Theoretically, g should be directly linked with the wavelength and the ice particle shapes, but as g is not measurable, we used constant values estimated by Libois et al. (2014), who combined simulations and *in situ* measurements of reflectance in Antarctica and the French Alps. They found that using B = 1.6 and g = 0.86 is more realistic to model snow optical properties rather than considering spherical

grains.

The albedo of a flat smooth surface obtained with ART ($\alpha_{flat}$) at wavelength $\lambda$ and at $\theta_s$ is deduced as follows:

$$\alpha_{flat}(\lambda, \theta_s) = r_{diff-tot}(\lambda, \theta_s)\alpha_{diff}(\lambda) + (1 - r_{diff-tot}(\lambda, \theta_s))\alpha_{dir}(\lambda, \theta_s) \hspace{1cm} [5],$$

where $r_{diff-tot}(\lambda, \theta_s)$ is the ratio of diffuse-to-total illumination at wavelength $\lambda$ and at $\theta_s$, measured in the field shortly after each albedo measurement.

These formulations apply to a strictly leveled terrain (better than 0.5°). To account for the slope and compute the apparent albedo of a titled smooth surface, called $\alpha_{sim,smooth}$, a K factor is applied (Dumont et al., 2017), such as:

$$K = cos(\theta_n) + tan(\theta_s)sin(\theta_n)cos(\varphi_s - \varphi_n) \hspace{2cm} [6],$$

and:

$$\alpha_{sim,smooth}(\lambda, \theta_s) = r_{diff-tot}(\lambda, \theta_s).\alpha_{diff}(\lambda) + (1 - r_{diff-tot}(\lambda, \theta_s))K\alpha_{dir}(\lambda, \sim\theta_s) \hspace{1cm} [7],$$

where $\widetilde{\theta}_s$ is the effective $\theta_s$ modified with the slope. As shown by Dumont et al. (2017), the K factor is the relative change in the cosine of the sun effective incident angle to the slope, and makes it possible to reproduce the distortion of the spectra due to the presence of the slope (with potential albedo values above 1 in the case of a sun-facing slope; Picard et al., 2020).

Following the ART theory, Kokhanovsky and Zege (2004) (further referred as the KZ04 approximations) estimated the snow BRDF distribution by calculating reflectance over a hemisphere with the reflection function of a semi-infinite medium:

$$R(\Phi, cos\theta_s, cos\theta_v) = R_0(\Phi, cos\theta_s, cos\theta_v)exp\left(\frac{-Ak_vk_s}{R_0}\right) \hspace{1cm} [8],$$

where the function $R_0(\Phi, cos\theta_s, cos\theta_v)$ is the reflection function at $\omega_0 = 1$ (Kokhanovsky, 2013), with $\omega_0$ the single scattering albedo. $\Phi$ is the relative azimuth angle, $cos\theta_v$ is the cosine of the viewing zenith angle, $cos\theta_s$ is the cosine of the solar zenith angle, and A is estimated as follows:

$$A = 4\sqrt{\frac{1-\omega_0}{3(1-g)}} \hspace{2cm} [9].$$

$k_s$ and $k_v$ are called the escape functions, and are given by Kokhanovsky (2003) as:

$$k_s = \frac{3}{7}(1 + 2cos\theta_s) \hspace{2cm} [10],$$

and:

$$k_v = \frac{3}{7}(1 + 2cos\theta_v) \hspace{2cm} [11].$$

## 3.2 Algorithms and model architecture

The Monte Carlo photon light transport algorithm propagates a large number of photons from their source to termination (i.e. that escape from the scene).

A photon is a particle of light carrying a flux and described by its power (intensity), its origin $\vec{r}$, and its propagation direction $\vec{\iota}$. Each photon starts its trajectory with an intensity equal to 1 (unitless quantity of energy), and a direction $\vec{\iota}$ described with

the couple ($\theta_s$, $\varphi_s$) given as input. Photons are either absorbed or reflected at each hit according to the facet albedo value
(Iwabuchi, 2006), that is estimated with the single scattering properties in case of the KZ04 configuration, or as a constant
snow reflectance in case of the Lambertian configuration. The algorithm works as follows.

A flow chart of a photon path as computed with RSRT is presented in Figure 4. This is computed in four main steps.

**Step 1: Estimate the next intersection of the photon with the mesh of the surface (called "hit").** The Bounding Volume
Hierarchies (BVH) technique (Ize, 2013) is used to efficiently search for the first facet in the photon propagation direction.
Basically, it uses a simple recursive intersection routine to test if the photon hits or does not hit the bounding volume, and
when positive, the hitting point is searched using a BVH algorithm (Wald et al., 2007). The precise intersection point within
the facet is determined by applying the watertight ray/triangle intersection algorithm (Woop et al., 2013). If the photon hits a
facet, its origin $\vec{r}$ is updated on the intersected facet. The normal of the facet is estimated. If there is no hit, the photon escapes
from the mesh, and depending on its direction (upward or downward), its intensity is added to the down or up welling radiation
bin (Fig. 4).

**Step 2: Update the intensity.** The photon intensity at hit $n$ (called $i_{p,n}$) is weighted by the spectral albedo accounting for the
incoming direction angles $\alpha_{flat}(\lambda, \theta_i)$ as follows:

$$i_{p,n+1} = i_{p,n}\,\alpha_{flat}(\lambda, \theta_i) \hspace{4cm} [12],$$

Two configurations are possible: With the KZ04 configuration, the hit facet is considered as a snow surface and $\alpha_{flat}(\lambda, \theta_i)$ is
estimated by considering the local incident angle $\theta_i$ and snow properties (SSA, B, g) (i.e. with ART, Eq. (5)). With the
Lambertian configuration, the hit facet is a Lambertian surface (i.e. isotropic diffusion), and the $\alpha_{flat}(\lambda, \theta_i)$ is a constant value
equal to $\alpha_{flat}(\lambda)$, given as an input of RSRT.

**Step 3: Sample the outgoing direction.** The most likely outgoing direction of the photon after a hit is estimated from the
BRDF distribution computed with the KZ04 approximations (Sect. 3.1). Thus, the next direction after the scattering depends
of the incident angle of the photon and snow properties. With the KZ04 approximation, the surface is more forward scattering
than for a Lambertian surface (Warren, 1982). BRDF values are estimated for all directions, defined by the ($\cos\theta_v$, $\varphi_v$) pair.
The outgoing direction is sampled from the BRDF distribution using a rejection algorithm as follows: in a first step, the azimuth
$\varphi_v$ is sampled from a uniform distribution between 0 and $2\pi$, and $\cos\theta_v$ with a uniform distribution between 0 and 1, so that the
hemisphere is sampled with a cosine weighting distribution (Greenwood, 2002). In a second step, a probability of acceptance
is given to each direction ($\theta_v$, $\varphi_v$). This probability of acceptance is estimated by the BRDF value in this direction, normalized
by the maximum value of the BRDF distribution.

**Step 4:** Update the direction $\vec{l}$. The new photon direction $\vec{l}_{n+1}$ after the hit n is updated as follows:

$$\vec{l}_{x,n+1} = \vec{l}_n - \cos\theta_i.\vec{n} \hspace{4cm} [13],$$

$$\vec{l}_{y,n+1} = \vec{n} \times \vec{l}_{x,n} \hspace{4cm} [14],$$

$$\vec{l}_{n+1} = \sin\theta_i(\vec{l}_{x,n+1}.\cos\varphi + \vec{l}_{y,n+1}.\sin\varphi) - (\vec{l}_n.n).\vec{n} \hspace{2cm} [15],$$

With $\vec{l}_{x,n}$ and $\vec{l}_{y,n}$ the photon directions in the x and y axis before the hit n, respectively.

The algorithm returns to step 1 until the photon escapes from the scene (Fig. 4), or until its intensity is lower than a threshold
(set to 0.01 in RSRT). To ensure an unbiased termination in the latter case, a 'Russian roulette' method is applied (Iwabuchi,
2006), which consists in: accepting or rejecting the termination with probabilities $1-p$ and $p$, respectively ($p = 0.2$ in RSRT).
In case of rejection, the weak intensity of the photon is rescaled by the factor $1/p$, and the algorithm goes again to step 1. As
explained by Iwabuchi (2006), the total energy is conserved for any $p$ value, and this approach can be applied at any step of
the algorithm.

At the end of its path, the photon intensity is counted in: 1) the total upward intensity ($I_\uparrow$) if the photon escapes with an
upward direction, 2) the intensity lost downward if its final z axis direction is downward (this is possible with a tilted surface
for instance). If the latter contribution is higher than $10^{-3}$ for a horizontal rough surface, we consider that too many photons

have been lost to output a realistic albedo, meaning that the simulation used a too wide radiation source or conversely a too small mesh area.

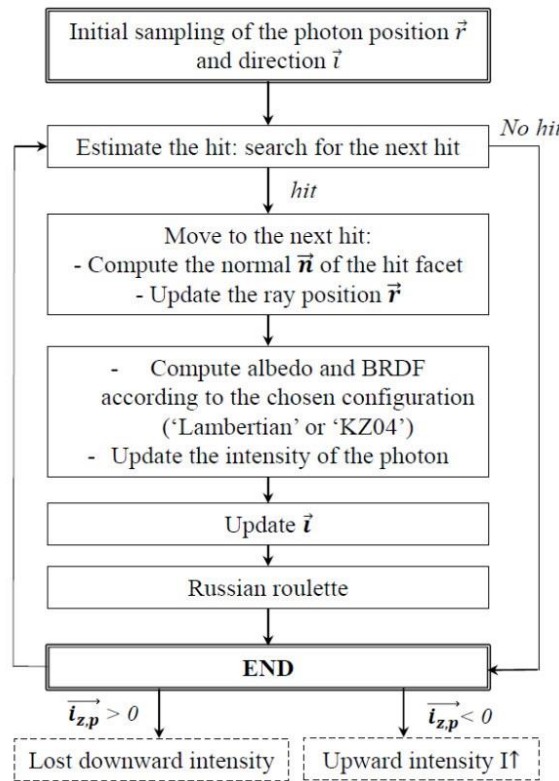

**Figure 4. Flow chart of a photon path in the RSRT algorithm. → $i$ is the incident direction of the photon, → $i_{z,p}$ is the z axis component of the photon at the end of its path.**

### 3.3 Simulation framework

#### 3.3.1 Model simulations

RSRT is run here either by considering the snow surface 1) as Lambertian (Lambertian configuration), the albedo is not sensitive to the incident angle and each photon hitting the mesh is reflected with a constant facet albedo equal to $\alpha_{flat}(\lambda)$, or 2) as a snow surface using KZ04 analytical equations (referred to as the KZ04 configuration). In this latter configuration, each photon hitting the mesh is reflected with $\alpha_{flat}(\lambda, \theta_i)$, which depends on the incident angle $\theta_i$, SSA, B and g values, i.e. by considering the intrinsic BRDF of the snow. The two configurations are compared in Sect. 4.4. The KZ04 configuration is used by default in for all other simulations to compare with observations.

RSRT inputs are described in Table 2. Triangular meshes of rough surfaces are modelled by reproducing same linear shapes as those created in the field with an orientation $\varphi_r$, a height $H$, a width $W$, and spaced by a constant distance defined with the period $\Lambda$ (as shown in Figure 1, with same values as in Table 1). Meshes have a spatial resolution of 1 cm and are produced large enough to be considered as infinite (no edge effects). When a RSRT simulation is started, an ensemble of photons is first created on a horizontal plane above the surface mesh and distributed quasi-randomly to produce a parallel source. The size of the photon ensemble is set to $10^6$ photons as a compromise between the computing time and a good representation of the emission source. The direction of propagation of the ensemble of photon is initialized with the solar zenith and azimuth angles given as inputs.

RSRT outputs the snow spectral albedo, either in direct or diffuse illumination conditions: $\alpha_{dir,rough}(\lambda, \theta_s)$ and $\alpha_{diff,rough}(\lambda)$, respectively, considering that the plane of the mesh is perfectly flat. Then, $\alpha_{dir,rough}(\lambda, \theta_s)$ and $\alpha_{diff,rough}(\lambda)$ are combined with Eqs. (6) and (7) to simulate the apparent snow albedo of a tilted rough surface, called $\alpha_{sim,rough}(\lambda, \theta_s)$, and therefore the simulated apparent albedo accounts for the slope characteristics and surface roughness. Each simulation assumes clear sky conditions,

and no atmosphere scattering and absorption is considered in the Monte Carlo algorithm. The only atmospheric parameter used in the model is the diffuse-to-total illumination ratio (which depends on atmospheric conditions). This parameter was measured in the field at each albedo acquisition (see Sect.2.1). At the small scale of this study, the effect of the atmosphere is negligible between the sensor and the surface. Future work should add the atmosphere in RSRT for applications over large-scale natural surfaces (mountainous areas).

**Table 2. RSRT inputs description**

| Inputs | Description | Units | Lambertian | KZ04 |
|--------|-------------|-------|------------|------|
| $\vartheta_s$ | Zenith angle of the radiation source | Degrees (clockwise) | x | x |
| $\varphi_s$ | Azimuth angle of the radiation source | Degrees (clockwise, 0° is North) | x | x |
| Mesh | Triangular mesh | With a 1 cm spatial resolution | x | x |
| z scale | Additional scaling coefficient of the mesh in the z axis. 1 is default, 0 to simulate a flat smooth surface | No units | x | x |
| $N_{photons}$ | Size of the photon ensemble | No units | x | x |
| $\varphi_r$ | Azimuthal orientation of the mesh around the z axis | Degrees (clockwise, 0° is North) | x | x |
| Facet albedo | Constant albedo $\alpha_{flat}(\lambda)$ | No units. By default = 0.8 | x | |
| B, g | Snow shape coefficients (Libois et al., 2014b) | No units | | x |
| SSA | Specific Surface Area of snow | $m^2\,kg^{-1}$ | | x |

### 3.3.2 Evaluation of simulations

The evaluation of simulations was treated over a set of N observations using the Root Mean Square Deviation (RMSD), defined as follow:

$$RMSD(\lambda, \theta_s) = \sqrt{\frac{\sum_{i=1}^{N}(\alpha_{sim,i}(\lambda,\theta_s) - \alpha_{obs,i}(\lambda,\theta_s))}{N}} \qquad [16]$$

with $\alpha_{sim,i}(\lambda, \theta_s)$ the i$^{th}$ simulation (either $\alpha_{sim,smooth}$ or $\alpha_{sim,rough}$) at the wavelength $\lambda$ and $\theta_s$, and $\alpha_{obs,i}(\lambda, \theta_s)$ the i$^{th}$ measured albedo. We further called $\overline{RMSD}$, the RMSD$(\lambda, \theta_s)$ averaged over the 600-1050 nm range for one spectrum.

The accuracies of $\alpha_{sim,smooth}(\lambda, \theta_s)$ and $\alpha_{sim,rough}(\lambda, \theta_s)$ are compared to evaluate the accuracy gain acquired by taking into account surface roughness. The main goal of this study is to quantify the roughness effect on albedo values and to determine if this effect is wavelength dependent. Therefore, statistical results are given at two wavelengths: one in the visible domain at 700 nm and one in the NIR domain at 1000 nm. The relation between roughness effect and SSA is investigated at 1000 nm since at this wavelength the albedo sensitivity to SSA is larger (Domine et al., 2006).

### 3.3.3 Impact of uncertainties

Albedo observations may have been affected by uncertainties or unmeasured variations in the field. To investigate the potential impact, we conducted the following simulations.

Firstly, SSA may have varied over time in the experiments A and B, whereas albedo was simulated with a constant SSA. In order to estimate these variations, we retrieved SSA at the beginning of the experiments from albedo observations over the smooth surfaces, by fitting $\alpha_{sim,smooth}$ with $\alpha_{obs}$ using the same approach as described by Libois et al. (2015). RSRT was then run by considering retrieved SSA values (SSA$_r$) for simulations over A-smooth and B-smooth-dry surfaces, and the measured SSA values (SSA$_m$) for simulations over the rough surfaces. Results are studied at 1000 nm where the albedo sensitivity to SSA is higher.

Secondly, the difference between retrieved and measured SSA may be related to the uncertainty in SSA measurements. We explored the impact of SSA uncertainties with RSRT simulations by varying $SSA_m$ by $\pm$ 10 % over the rough surfaces at 1000 nm.

Thirdly, the impact of slope uncertainties was studied with RSRT simulations by varying the slope of the rough surfaces by $\pm$ 1° in the experiments C rough 90° and D rough 90° at 1000 nm. We used C and D experiments only since observations over the rough and smooth surfaces were acquired simultaneously, with similar $\theta_s$ values (to not influence ~$\theta_s$ the effective $\theta_s$ modified with the slope).

### 3.3.4 Analysis of processes introduced by surface roughness

The variations of illumination conditions and SSA may attenuate or accentuate roughness effects by playing a role either in the effective angle effect, or in multiple reflections. We thus investigated separately these effects as a function of illumination conditions and SSA to better characterize roughness effects.

The effective angle effect is the alteration of the local incident angle over roughness shapes. It was simulated with RSRT using the KZ04 configuration (albedo varying with $\theta_s$) and by requiring that photons hit the surface only once, i.e. without multiple reflections. The total upward and downward intensities were then added to count all the photons that have not been absorbed after the first hit. We also conducted same simulations with the Lambertian configuration to check there was no angular dependence. These simulations were performed with various illumination conditions.

The effect of multiple reflections caused by the photon trapping depends on the albedo value. While the effective angle effect is significant under direct sunlight only, this second effect is significant both under direct and diffuse illumination (Warren et al., 1998). Therefore, it was simulated by running RSRT under diffuse sunlight. Simulations were conducted for various SSA.

### 4 Results and discussion

First, the new RSRT model is evaluated with *in situ* measurements (Section 4.1). Second, we explore the albedo sensitivity to macroscopic surface roughness through three questions: 1) is it possible to quantify the change in albedo caused by surface roughness, and to model this contribution (Sect. 4.2)? 2) What is the impact of SSA and slope uncertainties in the quantification of roughness effects (Sect. 4.3)? 3) What are the respective roles of the effective angle effect and multiple reflections according to snow properties and illumination conditions (Sect. 4.4)? The impact of roughness on the absorbed energy is also investigated (Sect. 4.5).

### 4.1 RSRT evaluation

Table 3 shows RMSDs of albedo simulated by considering or neglecting the presence of roughness *($\alpha_{sim,rough}$* and *$\alpha_{sim,smooth}$*, respectively) at 700 nm and 1000 nm for each experiment.

For experiments A and B (Sect. 2.1), *$\alpha_{sim,smooth}$* RMSD increases with the fraction of roughness features *($\eta = W/\Lambda$)*, and is higher at 1000 nm than at 700 nm. By considering roughness, the simulations are more accurate by about a factor 2 at 700 nm and 1000 nm compared to *$\alpha_{sim,smooth}$* (average *$\alpha_{sim,rough}$* RMSD of 0.02 at 700 nm and 1000 nm), which is significant.

Figure 5 shows measured and simulated spectral albedo acquired when the surface is smooth and when the fraction of roughness features is the largest *($\eta$ = 27%)* for experiments A and B. Both surfaces have a sun-facing slope (3.1° for experiment A and 3.6° for experiment B, see Table 1), so albedo values above 1 in the visible range are not surprising, as explained in Sect. 2.4. For both experiments, the *$\alpha_{obs}$* spectra is lower in presence of surface roughness than the spectra acquired over the smooth surface. Indeed, when the number of roughness shapes increases, more photons are trapped between concavities. The

420 photons have a larger probability to be absorbed (one probability at each hit) relative to a smooth surface (only one hit), causing the observed albedo to decrease.

The $\alpha_{sim,rough}$ spectra follow the observed trend. Simulations are improved compared to $\alpha_{sim,smooth}$ that neglects surface roughness ($\overline{RMSD} = 0.02$ when $\eta = 27\%$). For both experiments, the pattern of the measured spectra between 600 nm and 700 nm are probably led by the presence of impurities (not visible to the naked eye in the field). Previous studies showed that even a small
concentration of snow LAPs induces a drastic decrease of the albedo in the visible range (Warren, 1984; Dumont et al., 2017), and may explain why measurements and simulations differ in the 600-700nm range. Moreover, the spectra do not overlap perfectly in the NIR domain, but differences are below 0.01, and it is probably because of a small bias in SSA measurements (10% uncertainty). Overall, taking into account the measurement errors, $\alpha_{sim,rough}$ spectra reproduces the observed spectra well for both experiments and the RSRT model improves the spectral albedo simulations by accounting for roughness, compared
to those which neglect them (Fig. 5).

**Table 3. RMSD of $\alpha_{sim,smooth}$ and $\alpha_{sim,rough}$ at 700 nm and 1000 nm. RMSD is calculated with Eq. (16). N is the number of studied surfaces. For experiments C and D, RMSDs are calculated for the simulations over the rough surface.**

| | $\eta$ | $\lambda = 700$ nm | | $\lambda = 1000$ nm | |
|---|---|---|---|---|---|
| | | $\alpha_{sim,smooth}$ | $\alpha_{sim,rough}$ | $\alpha_{sim,smooth}$ | $\alpha_{sim,rough}$ |
| Experiments A and B (N=2) | 7 % | 0.02 | 0.01 | 0.03 | 0.01 |
| | 13 % | 0.03 | 0.02 | 0.05 | 0.02 |
| | 27 % | 0.04 | 0.02 | 0.09 | 0.03 |
| Experiments C and D (N = 19) | 63 % | 0.07 | 0.03 | 0.09 | 0.04 |
| Total (N = 21) | 7-63 % | 0.06 | 0.03 | 0.08 | 0.04 |

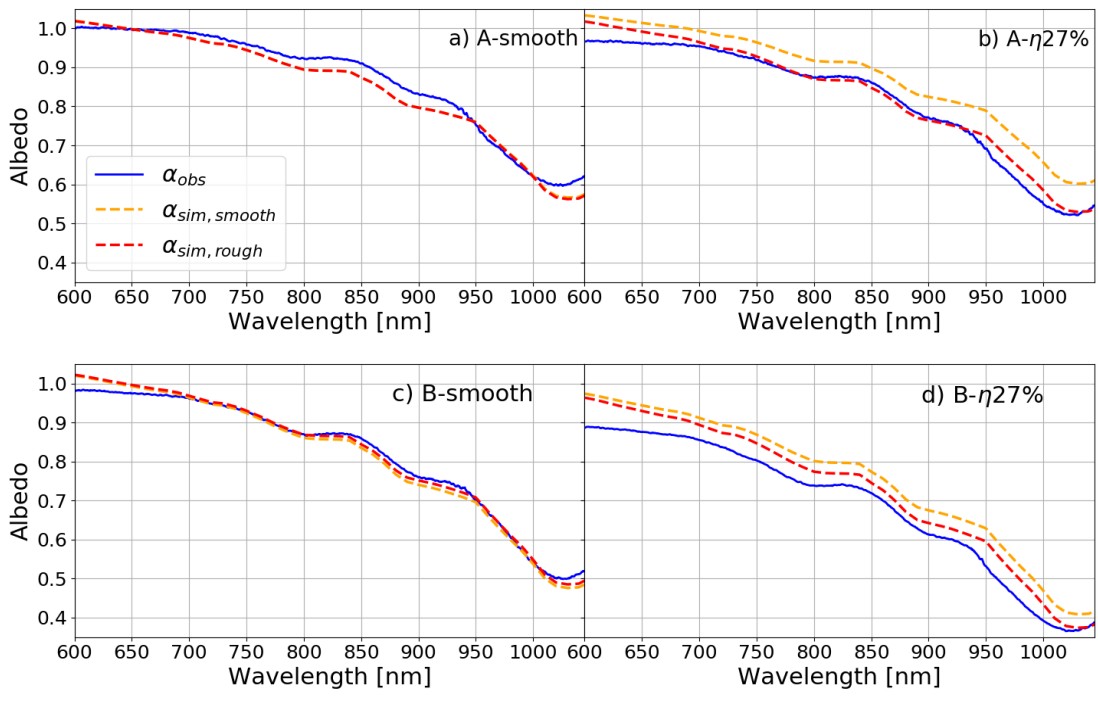

**Figure 5. Measured spectral albedo $\alpha_{obs}$ (blue full lines), and spectral albedo simulated with RSRT by considering the surface as smooth ($\alpha_{sim,smooth}$, orange dotted lines), and by considering surface roughness ($\alpha_{sim,rough}$, red dotted lines) for a) A-smooth, b) A-$\eta$27%, c) B-smooth, d) B-$\eta$27%.**

In Experiments C and D, albedo measurements are simultaneously acquired over a rough surface and a nearby smooth surface for multiple illumination conditions, every 20 minutes. Albedo simulations over the rough surface are significantly improved

by modelling surface roughness compared to those modelling a smooth surface: $\alpha_{sim,rough}$ has an averaged RMSD of 0.03 at 700 nm and 0.04 at 1000 nm (against an averaged RMSD of 0.07 at 700 nm and 0.09 at 1000 nm for $\alpha_{sim,smooth}$, Table 3).

To illustrate the spectral performance of the RSRT model, Figure 6 shows albedo measurements and simulations from 600 nm to 1000 nm for the experiments C at $\theta_s \sim 68°$ and D at $\theta_s \sim 59°$. For this example, randomly chosen illumination conditions were chosen for each experiment. For both experiments, the $\alpha_{obs}$ spectra show a significant decrease caused by the presence of surface roughness (~ -0.05 on average), more pronounced in the NIR domain (Fig. 6a and 6d). For Experiment C, the apparent albedo exceeds 1 in the visible range because of the presence of a sun-facing slope (3.3° - 4°, see Table 1).

By considering the surface roughness, the simulations are in agreement with observations, with small differences in the NIR domain maybe due to weak measured SSA uncertainties. The 0.05 decrease is reproduced well by the RSRT model when it accounts for surface roughness. This pattern is not reproduced at all by simulations considering the rough surface as smooth. For Experiment D, $\alpha_{sim,rough}$ spectra do not overlap the observations perfectly, though the decrease is followed (Fig 6d). This bias is due to several factors that are discussed further. Nevertheless, $\alpha_{sim,rough}$ simulations have a $\overline{RMSD}$ of 0.04 and are more accurate compared to simulations which do not take surface roughness into account ($\alpha_{sim,smooth}$), and which have a $\overline{RMSD}$ of 0.06.

Considering all observations, albedo simulations with the RSRT model are improved by a factor 2 by accounting for surface roughness ($\alpha_{sim,rough}$) at 700 nm and 1000 nm compared to those neglecting them ($\alpha_{sim,smooth}$), with an average RMSD of 0.03 at 700 nm and 0.04 at 1000 nm (Table 3). To the best of our knowledge, this is the first model capable of simulating spectral albedo taking into account the actual surface roughness, the topography and snow optical properties using a Monte Carlo photon transport algorithm.

Nevertheless, reductions in albedo due to roughness effects only are not clearly quantifiable here since several contributions change the albedo (illuminations and snow conditions also vary), and they have different impacts according to the frequency domain studied. The albedo sensitivity is thus further investigated at two wavelengths only, 700 nm and 1000 nm.

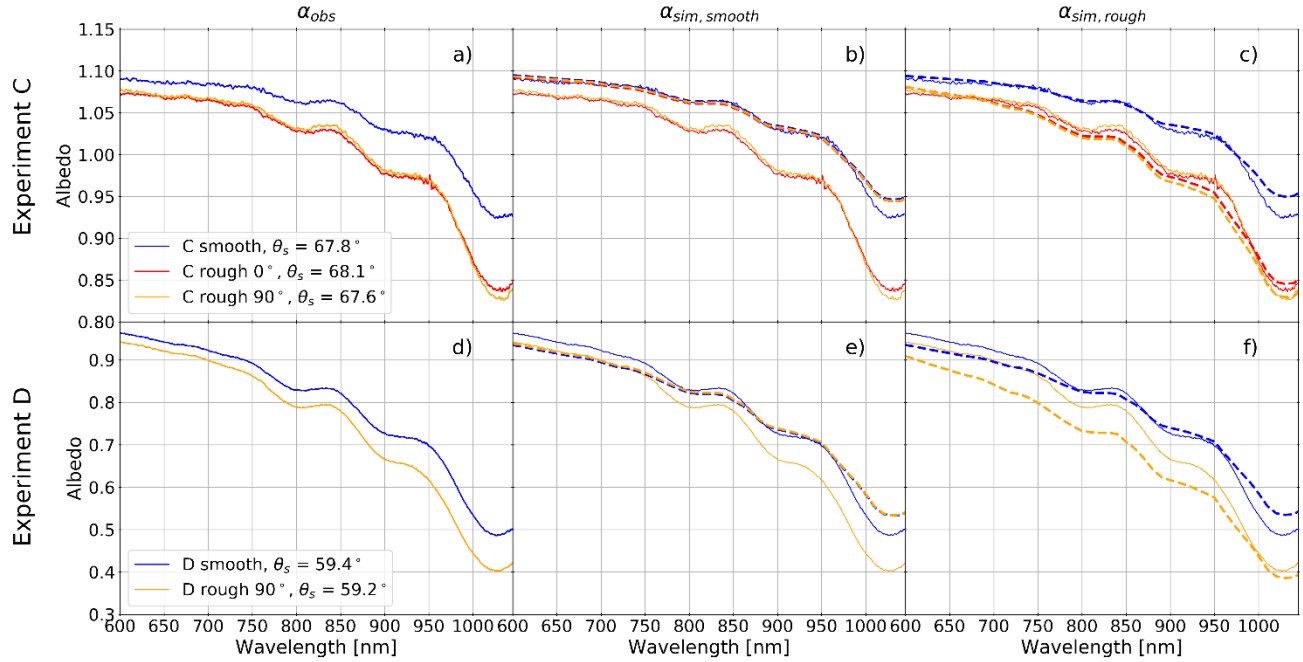

**Figure 6. Spectral albedo variations for experiment C at $\theta_s \sim 68°$ with a) $\alpha_{obs}$; b) $\alpha_{obs}$ (full lines) and $\alpha_{sim,smooth}$ (dotted lines), c) $\alpha_{obs}$ (full lines) and $\alpha_{sim,rough}$ (dotted lines). Red lines represent the C rough 0° surface, yellow lines the C rough 90° surface and blue lines the C smooth surface. Figures d), e) and f) are similar but for experiment D at $\theta_s \sim 59°$. Orange lines represent the D rough 0° surface and blue lines the D smooth surface.**

 **4.2 Albedo sensitivity to roughness features**

**4.2.1 Sensitivity to the fraction of roughness features**

To highlight the roughness effect, Figure 7 shows the change in albedo with increasing roughness fraction $\eta$ $(\eta = W/\Lambda)$ relative to the initial smooth surface, for both observations and simulations of experiments A and B, i.e. $\Delta\alpha_{obs}(\lambda, \theta_s) = \alpha_{obs}(\lambda, \theta_s) - \alpha_{obs}(\lambda, \theta_{s,o})$ and $\Delta\alpha_{sim,rough}(\lambda, \theta_s) = \alpha_{sim,rough}(\lambda, \theta_s) - \alpha_{sim,smooth}(\lambda, \theta_{s,o})$. However, this change in albedo is also affected by concomitant variations of the solar zenith angle $\theta_s$, as roughness features were added progressively to the initially smooth surface (Fig. 2). To quantify the impact of this spurious change, Fig. 7 also shows the simulated change in albedo if the surface had remained smooth $(\Delta\alpha_{sim,smooth}(\lambda, \theta_s) = \alpha_{sim,smooth}(\lambda, \theta_s) - \alpha_{sim,smooth}(\lambda, \theta_{s,o}))$.

In Experiment A, the stronger $\Delta\alpha_{obs}$ decrease is of 0.03 at 700 nm, and of 0.07 at 1000 nm, from A-smooth $(\eta = 0\ \%, \theta_{s,0} = 56.7°)$ to A-$\eta27\%$ $(\eta = 27\ \%, \theta_s = 63.6°)$ (Fig. 7a and 7c). Even a low fraction of roughness features $(\eta = 7\ \%)$ causes an albedo decrease of 0.02 compared to that of a smooth surface at 700 nm (and of 0.03 at 1000 nm). In theory, if the surface remained smooth throughout the experiment, $\alpha_{obs}$ should increase when $\theta_s$ increases (i.e the sun went up): photons penetrate less deeply into the snowpack as they enter with a grazing angle (large $\theta_s$). They encounter the first scattering event near the surface and have a larger probability to escape compared to a photon penetrating deeper with a low $\theta_s$ (Carroll and Fitch, 1981; Warren, 1982). By adding surface roughness, $\alpha_{obs}$ shows the inverse trend (Fig. 7a and 7b) and decreases with the increase of $\theta_s$, showing that albedo is more sensitive to roughness effects than to $\theta_s$ variations here. This result highlights the need to consider the presence of roughness in albedo simulations.

Simulations neglecting roughness follow the theory for a smooth surface, $\Delta\alpha_{sim,smooth}$ increases while $\theta_s$ becomes larger, by 0.02 at 700 nm, and by 0.03 at 1000 nm, between A-smooth and A-$\eta27\%$ (Fig. 7a and 7c). Simulations considering roughness follow the observation trend, $\Delta\alpha_{sim,rough}$ decreases by 0.01 at 700 nm, and by 0.03 at 1000 nm between A-smooth and A-$\eta27\%$. Nevertheless, RSRT (i.e. $\alpha_{sim,rough}$) underestimates by almost a factor 2 the observed albedo reduction. The reason of this underestimation may be linked to the SSA variations throughout the experiment.

In Experiment B, $\Delta\alpha_{obs}$ shows a strong decrease of 0.11 at 700 nm, and 0.15 at 1000 nm, between B-smooth-dry $(\eta = 0\ \%, \theta_s = 55.4°)$ and B-$\eta27\%$ $(\eta = 27\ \%, \theta_s = 39.9°)$ (Fig. 7b and 7d). In this experiment, $\alpha_{obs}$ decreases due both to the $\theta_s$ decrease (the sun went up, Sect. 2.1) and the $\eta$ increase. To remove $\theta_s$ contribution, we use the $\Delta\alpha_{sim,smooth}$ trend that depends on $\theta_s$ variations only: $\Delta\alpha_{sim,smooth}$ lowers when $\theta_s$ decreases and the reduction is half of that of $\Delta\alpha_{obs}$ (Fig. 7). In other words, half of the $\alpha_{obs}$ decrease is attributable to the decrease in $\theta_s$, and the other half to the presence of roughness. More precisely, by calculating $\Delta\alpha_{obs} - \Delta\alpha_{sim,smooth}$ we quantify the roughness effect on the albedo. The presence of roughness lowers the albedo of 0.06 at 700 nm and of 0.08 at 1000 nm when $\eta = 27\%$.

$\Delta\alpha_{sim,rough}$ decreases by 0.07 at 700 nm, and 0.11 at 1000 nm, between B-smooth-dry and B-$\eta27\%$ (Fig. 7b and 7d). Simulations are consistent with observations by considering the presence of roughness, but the simulated decrease is still underestimated compared to measurements, as for experiment A.

To accurately quantify roughness effects on albedo, it is important to compare rough and smooth surfaces for similar snow and illumination conditions. This is why we simultaneously measured albedo over B-$\eta27\%$ $(\eta = 27\%, \theta_s = 39.9°)$ and a nearby smooth surface (the B-smooth-wet surface: $\eta = 0\%, \theta_s = 36.4°$, Table 1 and Fig. 7b and 7d). The concurrent measurements show a decrease by 0.05 at 700 nm, and 0.07 at 1000 nm. This reduction is solely attributable to the presence of roughness. It is similar to the $\Delta\alpha_{obs}$ decrease by subtracting the $\Delta\alpha_{sim,smooth}$ that is caused by the $\theta_s$ decrease only (Fig. 7).

For both experiments, observations show that the albedo decrease is stronger when 1) the number of roughness features is larger, and 2) at the longer wavelengths. As albedo is lower in the NIR domain, the impact of multiple reflections is stronger. Indeed, the effect of multiple reflection is more important for intermediate values than for albedo close to 0 or 1 (i.e. systematic absorption or reflection, Warren et al., 1998).

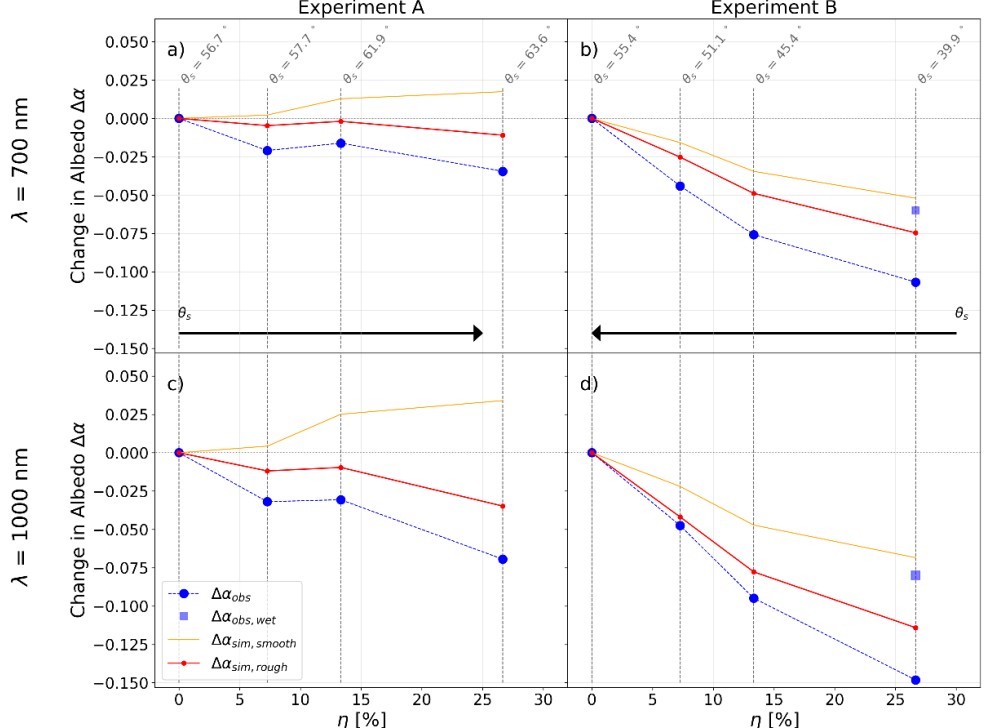

**Figure 7. Variations of albedo differences between the albedo at $\theta_s$ and the albedo at $\theta_{s,o}$, corresponding to that of the smooth surface, as a function of the $\eta$ ratio (W/Λ in %). Blue points are $\Delta\alpha_{obs}(\lambda, \theta_s)$ ($=\alpha_{obs}(\lambda, \theta_s) - \alpha_{obs}(\lambda, \theta_{s,o})$), orange lines are $\Delta\alpha_{sim,smooth}$ ($=\alpha_{sim,smooth}(\lambda, \theta_s) - \alpha_{sim,smooth}(\lambda, \theta_{s,o})$), where variations are due to $\theta_s$ changes only ($\eta=0\%$ for all simulations), and red lines are $\Delta\alpha_{sim,rough}$ ($=\alpha_{sim,rough}(\lambda, \theta_s) - \alpha_{sim,smooth}(\lambda, \theta_{s,o})$), which varies with $\eta$ and $\theta_s$. Blue squares are the $\Delta\alpha_{obs,wet}(\lambda, \theta_s)$ ($=\alpha_{obs}(\lambda, \theta_s) - \alpha_{obs-wet}(\lambda, \theta_{s,o})$) where $\alpha_{obs,wet}$ is the measured albedo over the B-smooth-wet surface ($\eta=0\%$ and $\theta_s=39.9^\bullet$, Table 1). Grey vertical lines describe the solar zenith angle ($\theta_s$) when measurements were acquired. Results are given for a) experiment A at 700 nm; b) Same as a) but for experiment B; c) experiment A at 1000 nm; d) Same as c) but for experiment B.**

### 4.2.2 Sensitivity to the roughness orientation

The albedo sensitivity to the roughness orientation with respect to the solar azimuthal angle ($\Delta\varphi_r$) is investigated at 700 nm and 1000 nm with experiments C and D, where measurements are simultaneously acquired over a smooth and a rough surface. Figure 8 shows the change in albedo as a function of $\Delta\varphi_r$ for both wavelengths. $\Delta\alpha_{obs}$ is the difference between $\alpha_{obs}$ acquired over the rough and the smooth surfaces at the same moment. Similarly, $\Delta\alpha_{sim,rough}$ is the difference between $\alpha_{sim,rough}$ simulated over the rough surface and $\alpha_{sim,smooth}$ simulated over the smooth surface, at same illumination conditions. Thus, the change in albedo is not correlated to $\theta_s$ here, but to $\varphi_s$ that leads the roughness orientation with respect to the sun position.

When roughness features are parallel to the sun (i.e. when $\Delta\varphi_r = 0°$ in Fig. 8a and 8d), $\alpha_{obs}$ decreases of 0.01 at 700 nm, and of 0.08 at 1000 nm, relative to a smooth surface. The impact becomes larger when the roughness orientation is perpendicular to the sun (when $\Delta\varphi_r = 90°$ in Fig. 8b and 8e), with an $\alpha_{obs}$ decrease of 0.02 at 700 nm and of 0.10 at 1000 nm. Thus, for experiment C, the reduction in albedo is 20 % stronger when roughness features lie perpendicular to the sun than when they are parallel. This is explaining by the fact that, when the sun elevation is low, if the roughness orientation is perpendicular to the sun, the effective incident angle over sides facing the sun is decreased compared to that of a smooth surface. In addition, the fraction of shadow is higher when $\Delta\varphi_r = 90°$. This effective angle effect leads to an average decrease in snow albedo relative to a smooth surface. However, for the C rough 90° experiment (Fig. 8b and 8e), $\Delta\varphi_r$ varies from 50° to 122° and $\Delta\alpha_{obs}$ does not show a strongest albedo reduction around 90°. Similarly, for C rough 0° (Fig. 8a, and 8d), $\Delta\alpha_{obs}$ values were not symmetrical to $\Delta\varphi_r = 0°$. This is caused by others contributions that are added to the roughness effects. First, the effect of the slope on albedo varies over time with the solar angle changes. Here we selected a smooth surface with a similar slope to that of the rough surface, so as to minimize the contribution of the slope by comparing rough-smooth albedo at similar illumination

conditions ($\Delta\alpha_{obs}$). The slope sensitivity to roughness effects is studied in Section 4.3.2. Second, the particularly high values
of SSA for this experiment ($\sim$100 m$^2$ kg$^{-1}$) induces lower absorption (Warren et al., 1998), and it may explain the albedo insensitivity to small variations of roughness orientation. Moreover, instead of a clear dependence between $\Delta\alpha_{obs}$ and $\Delta\varphi_r$, $\Delta\alpha_{obs}$ pattern shows oscillations, probably caused by the small differences in snow properties between the smooth and the rough surfaces. Indeed, SSA values over the smooth surface are homogeneous, while SSA values over the rough surface evolve unevenly according to the illumination received in the concavities during the day. The SSA sensitivity to roughness effect on
albedo measurements is investigated in Section 4.3.1.

In Experiment C, $\Delta\alpha_{sim,rough}$ variations reproduce well the $\Delta\alpha_{obs}$ decrease, with the same order of magnitude: the average decrease is of 0.01 at 700 nm and 0.08 at 1000 nm for C rough 0°, and of 0.02 at 700 nm and 0.10 at 1000 nm for C rough 90° (Fig. 8a and 8d).

In Experiment D rough 90°, measurements were acquired in morning, so $\Delta\varphi_r$ varies from 42° to 72° (Fig. 8c and 8f). We
measured an average $\Delta\alpha_{obs}$ decrease of 0.02 at 700 nm and 0.09 at 1000 nm, which is in agreement with results found for C rough 90°. In Fig. 8c and 8f, the $\Delta\alpha_{obs}$ increases when $\Delta\varphi_r$ goes from 42° to 72°, while in theory it should decrease when $\Delta\varphi_r$ approaches 90°. A possible explanation is that melting was observed at the surface in the field, resulting in a smoothing of our roughness shapes during the day, which attenuates the roughness effect on albedo values. Therefore, we cannot conclude on this observed trend since several contributions drove the measured albedo.

Fig. 8c and 8f shows that $\alpha_{sim,rough}$ overestimates by almost a factor 2 the reduction in $\alpha_{obs}$: the average $\Delta\alpha_{sim,rough}$ decrease is of 0.06 at 700 nm, and of 0.15 at 1000 nm. By considering roughness shapes constant along the day, $\Delta\alpha_{sim,rough}$ decreases when $\Delta\varphi_r$ goes from 42° to 72° (i.e. $\Delta\varphi_r$ gets closer to 90°). This trend is coherent with the theory, but more in situ measurements are needed to fully quantify the dependence of the apparent albedo to the roughness orientation.

To sum up, observations show that an increase of the number of roughness features leads to a larger reduction in $\alpha_{obs}$, with a higher sensitivity in the NIR domain. Roughness effects are also larger when the roughness orientation is perpendicular to the sun rather than parallel. $\alpha_{sim,rough}$ shows an overestimation of the observed albedo decrease, but observations may have been affected by uncertainties or unmeasured variations.

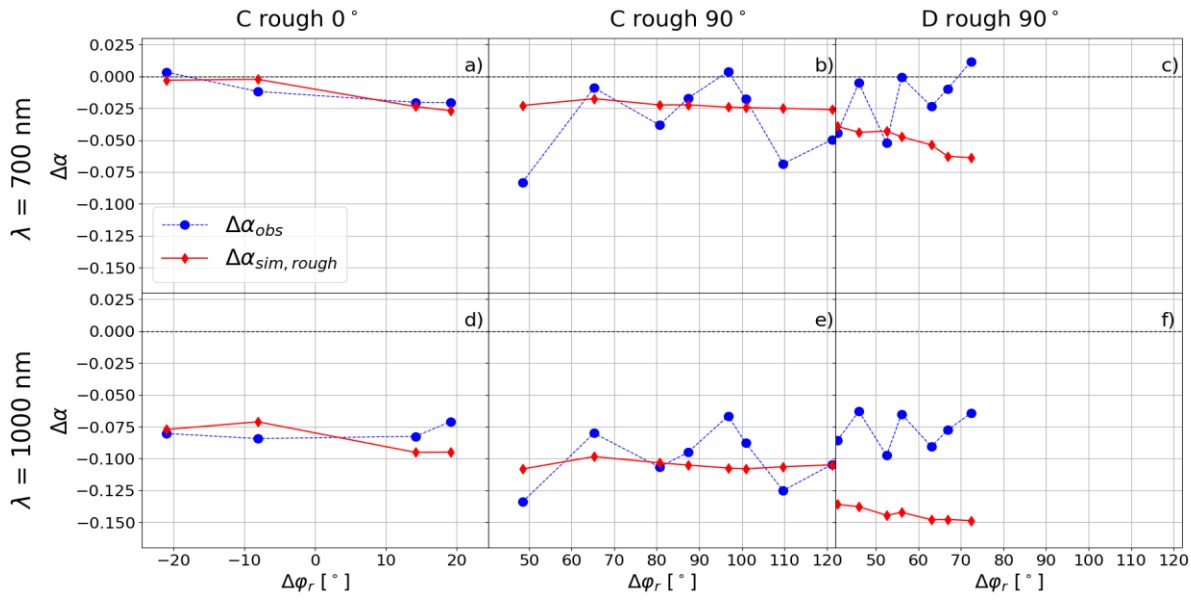

**Figure 8. Measured and simulated variations of Δα (=[rough-smooth] at the same $\theta_s$) at 700 nm as a function of $\Delta\varphi_r$ for a) the C rough 0° experiment, b) the C rough 90° experiment, c) the D rough 90° experiment. d), e) and f) are the same but at 1000 nm. Blue points are $\Delta\alpha_{obs}$, and red lines with diamonds are $\Delta\alpha_{sim,rough}$. The horizontal black dotted lines show the 0.**


**4.3 Analysis of uncertainties**

In a first step, we explore the possible SSA variations in the experiments A and B, and the impact on snow albedo. In a second step, we integrate SSA and slope uncertainties in our roughness analysis.

**4.3.1 Sensitivity to SSA**

    We estimate an SSA (written $SSA_r$) of 9.4 $m^2$ $kg^{-1}$ over A-smooth and 5.3 $m^2$ $kg^{-1}$ over B-smooth by fitting $\alpha_{sim,smooth}$ and $\alpha_{obs}$ (see Sect. 3.3.3 for the methodology). Measured SSA ($SSA_m$) are equal to 7.4 $m^2$ $kg^{-1}$ over A-$\eta13\%$ and 4.5 $m^2$ $kg^{-1}$ over B-

*$\eta13\%$.* Hence, for both experiments, there is a decrease in SSA from the beginning (smooth surface, *$\eta=0\%$) to $\eta = 13\%$,* which is compatible with the observation of melt at the surface during these two experiments performed in April. Indeed, Grenfell and Maykut (1977) explained that snow albedo decreases when liquid water replaces air between ice grains, and as the refractive index of the water is very close to that of ice, this results in an increase of the effective grain size (i.e a decrease of SSA).

To explore the impact of a decreasing SSA on albedo, RSRT is run by considering $SSA_r$ for simulations over A-smooth and B-smooth-dry surfaces ($SSA_r$ equal to 9.4 $m^2$ $kg^{-1}$ and 5.3 $m^2$ $kg^{-1}$, respectively), and $SSA_m$ for simulations over the rough surfaces (from *$\eta = 7\%$ to 27%,* $SSA_m$ equal to 7.4 $m^2$ $kg^{-1}$ and 4.5 $m^2$ $kg^{-1}$, see Table 1). Results are presented in Figure 9a and 9b, where *$\Delta\alpha_{sim,rough,ssa}$* is the difference $\alpha_{sim,rough}(\lambda,\ \theta_s,\ SSA_m) - \alpha_{sim,smooth}(\lambda,\ \theta_{s,o},\ SSA_r)$. Compared to *$\Delta\alpha_{sim,rough}$* (i.e. constant SSA), the *$\Delta\alpha_{sim,rough,ssa}$* decrease is multiplied by a factor two by considering both the increase in the fraction of roughness features,

and the SSA decline, from 9.4 to 7.4 $m^2$ $kg^{-1}$ (-15 %) for experiment A, or from 5.3 to 4.5 $m^2$ $kg^{-1}$ (-21 %) for experiment B (Fig. 9a and 9b). *$\Delta\alpha_{sim,rough,ssa}$* reproduces well the *$\Delta\alpha_{obs}$* decrease, with the same order of magnitude. Thus, the use of a constant SSA for $\alpha_{sim,rough}$ simulations in the experiments A and B probably explains the underestimation of the albedo reduction due to the presence of surface roughness and observed in Sect 4.2.1. Both SSA variations and roughness effects overlap and lower snow albedo in these two experiments, making it difficult to accurately isolate roughness effects.

Differences between retrieved and measured SSA may be explained by the uncertainty in SSA measurements (~ 10%, Arnaud et al., 2011). The impact of SSA uncertainties is investigated by varying SSA by ± 10 % in RSRT $\alpha_{sim,rough}$ simulations for all experiments. Obtained values range within the grey shade shown in Figures 9. Experiment C has large measured SSAs (~ 100 $m^2$ $kg^{-1}$), typical of fresh fallen snow, and SSA uncertainties affect slightly *$\Delta\alpha_{sim,rough}$* (Fig. 9c). On the contrary, a variation of ± 10 % in SSA strongly impacts the experiments with low SSAs: *$\Delta\alpha_{sim,rough,ssa}$* varies between 0.05-0.10 in the experiment A-

*$\eta27\%$* (Fig. 9a), between 0.11-0.16 in the experiment B-*$\eta27\%$* (Fig. 9b), and between 0.13-0.18 in the experiment D when *$\Delta\varphi_r$* = 72° (Fig. 9d). The reduction in albedo is stronger when SSA is lower due to higher absorption. More precisely, the grains at the surface control the first scattering event and large-coarse grains (i.e. low SSA) are both more absorptive and more forward scattering relative to fine grains since photons have to pass through longer paths in ice before being potentially scattered at the ice-air interfaces (Warren et al., 1998). Domine et al. (2006) have shown that the SSA-albedo relationship is non-linear and

that albedo varies slightly in the NIR domain when SSA > 30 $m^2$ $kg^{-1}$, while it is highly sensitive to SSA variations for SSA values below 10 $m^2$ $kg^{-1}$. Hence, in presence of surface roughness, a large SSA leads to a weaker impact of multiple reflections (high albedo), while the impact of the photon trapping is more important at low SSA (<10 $m^2$ $kg^{-1}$). There is a strong and nonlinear relationship between the roughness effect on the snow albedo and SSA values.

    Moreover, experiment D highlights that the impact of SSA uncertainties in albedo is linked to the roughness orientation (Fig.

9d). Albedo is twice as sensitive to SSA when *$\Delta\varphi_r$* = 72° as when *$\Delta\varphi_r$* = 42°. This is caused by the effective angle effect introduced by roughness: photons penetrate deeper over sides facing the sun when the roughness orientation is perpendicular to the sun (lower incident angle) than if it was oblique or parallel. When SSA is low, absorptions increase and a photon has larger probability to be absorbed by penetrating deeply in the snowpack. Hence, the effective angle effect is more pronounced when roughness orientation is perpendicular to the sun and for low SSA.

The joint impact of roughness effects and SSA in the NIR domain has consequences on the accuracy of SSA retrievals. Several studies directly used the ART equations to retrieve SSA from spectral albedo observations in the NIR (Dominé et al., 2006; Gallet et al., 2011; Libois et al., 2015; Picard et al., 2016). By neglecting roughness, SSA retrievals are underestimated to compensate for the albedo reduction caused by the presence of roughness. We retrieved SSAs for experiments C and D at each $\Delta\varphi_r$ by fitting $\alpha_{sim,smooth}$ and $\alpha_{obs}$ acquired over the rough surfaces (not shown). Compared to measured SSAs taken over the smooth surface, results demonstrate that roughness introduces a significant underestimation of the retrieved SSA, reaching 21 % for the roughness features considered here. Thus, it is important to use a model considering roughness to retrieve accurate SSA from albedo observations.

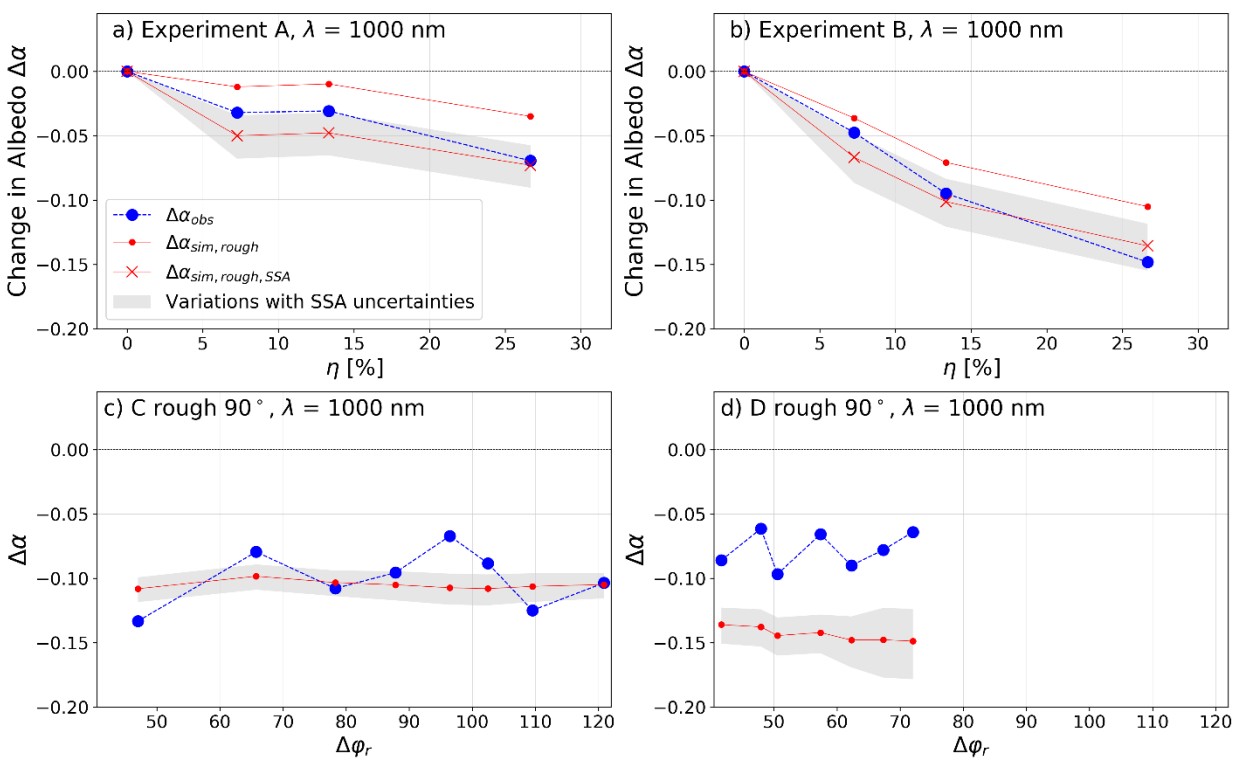

**Figure 9.** a) and b) are changes in albedo as a function of the $\eta$ ratio at 1000 nm for experiments A and B, respectively. Blue dotted lines are $\Delta\alpha_{obs}$ ($\alpha_{obs}(\lambda, \theta_s) - \alpha_{obs}(\lambda, \theta_{s,o})$). Red dotted lines with points are $\Delta\alpha_{sim,rough}$ ($\alpha_{sim,rough}(\lambda, \theta_s) - \alpha_{sim,smooth}(\lambda, \theta_{s,o})$) obtained using a constant SSA in RSRT (7.4 m² kg⁻¹ for A and 4.5 m² kg⁻¹ for B). Red lines with crosses are $\Delta\alpha_{sim,rough,ssa}$ ($\alpha_{sim,rough}(\lambda, \theta_s, SSA_r) - \alpha_{sim,smooth}(\lambda, \theta_{s,o}, SSA_m)$) obtained using SSA$_r$ for the smooth surface (at $\eta$ = 0 %, SSA$_r$ = 9.4 m² kg⁻¹ for A and 5.3 m² kg⁻¹ for B) and SSA$_m$ for rough surfaces ($\eta$ from 7 % to 27 %, SSA$_m$ = 7.4 m² kg⁻¹ for A and 4.5 m² kg⁻¹ for B). c) and d) are variations of $\Delta\alpha$ with $\Delta\varphi_r$ at 1000 nm for experiments C and D, respectively (similar to Figures 7e and 7f). $\Delta\alpha_{obs}$ and $\Delta\alpha_{sim,rough}$ are the observed and simulated albedo differences between the rough and the smooth surfaces at $\Delta\varphi_r$. Grey shades represent the range of $\Delta\alpha$ obtained by varying the SSA by ± 10% in RSRT simulations.

### 4.3.2 Sensitivity to the surface slope

The impact of slope uncertainties is explored by varying the slope by ± 1° for simulations over the rough surfaces for experiments C and D at 1000 nm (Sect. 3.3). Obtained values range within the grey shade shown in Figure 10. The albedo sensitivity to the slope depends of the slope aspect $\varphi_n$, with respect to the solar azimuthal angle $\varphi_s$, since the aspect controls the change in the incident angle ($\sim\theta_s$) relative to $\Theta_s$. The slopes have no impacts on albedo if the slope aspect is perpendicular to the solar azimuthal angle ([$\varphi_s - \varphi_n$] = 90° or 270°) since it has no effect on the solar incident angle. On the other hand, impacts change rapidly when the aspect $\varphi_n$ becomes parallel to $\varphi_s$ ([$\varphi_s - \varphi_n$] = 0° or 180°), as it is shown using Eq. (6) and (7). Over a titled rough surface with roughness orientation perpendicular to the sun ($\Delta\varphi_r$ = 90°) and a slope direction opposite to that of the sun ($\varphi_s - \varphi_n$ = 180°), roughness sides facing the sun experience a lower effective incident angle relative to a flat

rough surface, leading to a lower albedo. Fig. 10b illustrates this point for experiment D rough 90°: the albedo sensitivity is twice as strong when the slope direction is closer to 180° ($[\varphi_s - \varphi_n] = -150°$, i.e a slope opposite to that of the sun) than when it gets closer to 90° ($[\varphi_s - \varphi_n] = -120°$). Note that this experiment has low SSA, leading to a strong sensitivity to a change of the incident angle, as explained in previous section. Therefore, for low SSA, the impact of roughness on albedo is accentuated when the slope direction is opposite to the sun, and attenuated when the slope is facing the sun.

In Experiment C (Fig. 10a), the albedo is highly sensitive to slope uncertainties (variations of 0.05-0.15). However, due to high SSA there is a low albedo sensitivity to the $\varphi_s - \varphi_n$ angle (the effective angle effect is negligible). Therefore, the observed albedo sensitivity may be explained by a larger effect of multiple reflections, accentuated by the fact that $\Theta_s$ is particularly large for this experiment ($> 60°$).

To sum up, we have shown that the albedo sensitivity to roughness is larger when the SSA is low ($< 10$ m$^2$ kg$^{-1}$), when roughness features are perpendicular to the sun, and when the surface slope aspect is facing away the sun. The roughness effect if strongly linked to SSA values which affect: 1) the impact of the effective angle effect, since the decrease of the incident angle on roughness sides facing the sun has more consequences on the albedo when SSA is low (high absorption), 2) the impact of multiple reflections, which is larger when the probability of a photon to be absorbed or reflected is well balanced. To accurately quantify roughness effects, it is crucial to measure SSA regularly (a small variation may overlap the roughness effects) and to determine the slope. In our experiments C and D, where SSA was measured at each albedo acquisition, we have shown that even considering uncertainties of $\pm 10$ % of SSA and of $\pm 1°$ of slopes, roughness effects are significant and cause at least an albedo decrease of 0.06 in the experiment C rough 90°, and of 0.11 in the experiment D rough 90°, at 1000 nm.

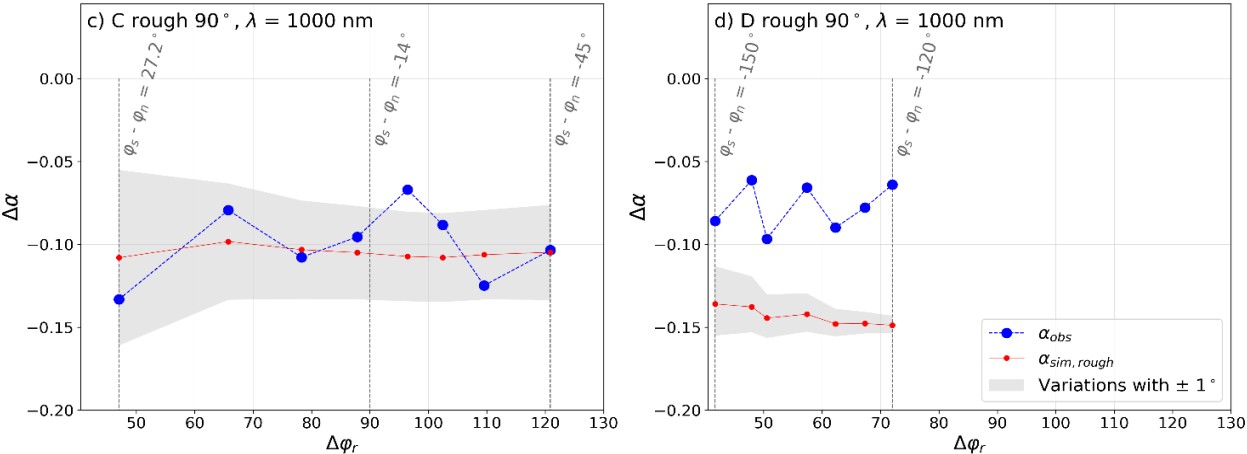

**Figure 10. Same as Figure 9c and 9d, except that the grey shades represent the range of $\Delta\alpha$ obtained by varying the slope by $\pm 1°$ in RSRT simulations, for a) experiment C rough 90°, and b) D rough 90°. $\varphi_n$ is the aspect of the slope and $\varphi_s$ is the solar azimuthal angle, separately given in Table 1. Vertical black lines indicates [$\varphi_s$ - $\varphi_n$] angles at the beginning and at the end of experiments (and at $\Delta\varphi_r$ =90° for experiment C).**

## 4.4 Analysis of the two roughness effects

The two processes introduced by surface roughness are decoupled using RSRT to better characterise roughness effects as a function of snow properties and illumination conditions.

### 4.4.1 Effective angle effect

To simulate the effective angle effect, we count all photons that have not been absorbed after the first hit. RSRT is run at 1000 nm with a KZ04 configuration, and we sum the total upward and downward intensity considering one hit only.

Simulations are performed for various $\theta_s$ and $\varDelta\varphi_r$. The initial conditions of the A-η27% experiment without slope and under

direct sunlight are used. Roughness shapes are rectangular and the SSA is low (7.4 $m^2 kg^{-1}$), which lead to a maximal effect of incident angle variations.

Figures 11a and 11b show the simulated $\varDelta\alpha_{sim,rough}$ ([rough-smooth] with similar illumination) as a function of $\theta_s$ and $\varDelta\varphi_r$. The Lambertian configuration yields a constant albedo, as expected since there is no incident angle dependence. The albedo decreases of 0.04 is due to shadow areas introduced by roughness features and that receive less radiations.

With the KZ04 configuration, Fig. 11a and 11b shows that the effective angle effect is strongly linked to illumination conditions. Firstly, as previously observed, the model predicts a strong drop in albedo when $\theta_s$ increases (Fig. 11a). When $\theta_s >$ 50°, with the rectangular roughness shapes of the experiment A, the local incident angle of photons hitting the vertical sides facing the sun is lower than that of a smooth surface when $\theta_s > 45°$ if $\varDelta\varphi_r = 90°$. Thus, photons penetrate deeply in the snowpack before being eventually redirected upward, which conduces to a stronger decrease in albedo relative to a smooth surface.

Conversely, when $\theta_s < 50°$, the effective incident angle is higher over roughness sides facing the sun compared to that of a smooth surface. It leads to an increase in albedo, and this is why $\varDelta\alpha$ is higher with the KZ04 configuration than with the Lambertian configuration when $\theta_s < 50°$ (Fig. 11a). Hence, the reduction in albedo depends on the slope of roughness sides (i.e. their shapes). Fig. 11a also illustrates that in the presence of roughness, albedo decreases more rapidly with $\theta_s$ at large values of $\theta_s$. Therefore, surface roughness plays a more important role at grazing angle (large $\theta_s$). Moreover, our results show

that the effects of roughness become negligible at 1000 nm when $\theta_s < 30°$. The albedo decrease caused by the effective angle effect only is of 0.04 for experiment A-η27%, when $\theta_s = 63°$ (Fig. 11a, [Lambert – KZ04]). Secondly, by changing the incidence angle, the roughness orientation also plays an important role (Fig. 11b). The reduction in albedo caused by the effective angle effect goes from 0 when $\varDelta\varphi_r = 0°$ to 0.09 when $\varDelta\varphi_r = 90°$ for experiment A.

To sum up, the albedo decrease due to the effective angle effect becomes rapidly stronger with $\theta_s$ at large $\theta_s$ ($\theta_s > 50°$) and

when $\varDelta\varphi_r = 90°$. In Experiment A, the model predicts a decrease in albedo of 0.07 when $\theta_s = 80°$ ([Lambert – KZ04] on Fig. 11a), caused by the effective angle effect only, i.e a drop 75% stronger compared to that of $\theta_s = 63°$. Therefore, it is necessary to account for the intrinsic BRDF of the snow to simulate realistic albedo over rough surfaces, in particular in Polar Regions where $\theta_s$ is high.

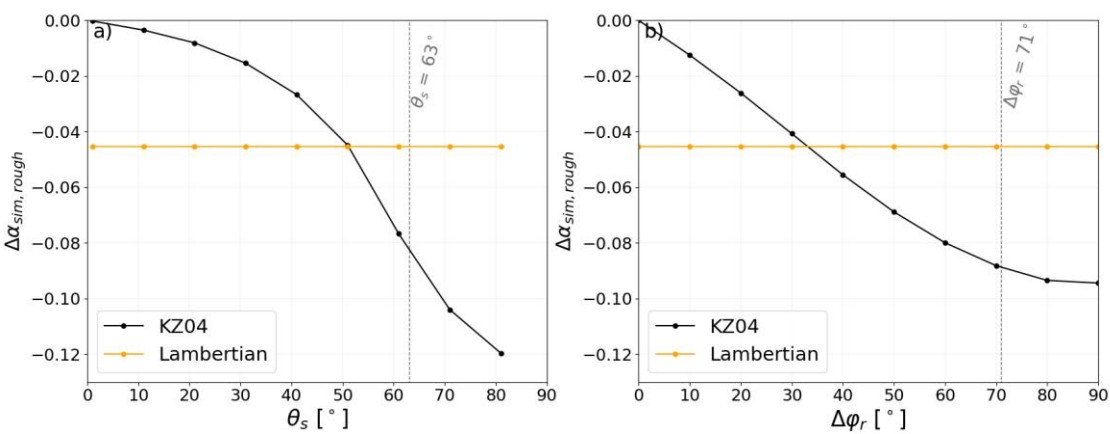


**Figure 11. Variations of $\varDelta\alpha_{sim,rough}$ ([rough-smooth] at same illumination) simulated with RSRT at 1000 nm with the initial condition of the experiment A-η27%, without slope as a function of a) $\theta_s$ (in degrees) and a constant $\varDelta\varphi_r = 71°$; and b) $\varDelta\varphi_r$ and a constant $\theta_s =$ 63°. Simulations are performed with the Lambertian configuration (in orange) and the KZ04 configuration (in black). Vertical dotted lines indicate the initial condition of the experiment A-η27% (Table 1).**

### 4.4.2 Multiple reflections


RSRT is run by varying SSA values with the KZ04 configuration and under diffuse sunlight to simulate the trapping effect of photons only (for the A-$\eta$27% experiment, *see* Sect. 3.3 for details). Simulations are performed over a smooth and a rough surface to compute $\Delta\alpha_{sim,rough}$. Results are shown in Figure 12 as a function of SSA. The impact of multiple reflections is higher for SSA between 8 m² kg⁻¹ and 14 m² kg⁻¹, with a maximum effect at SSA = 9 m² kg⁻¹. For the experiment A-$\eta$27%, the measured

SSA is of 7.4 m² kg⁻¹, and it induces a simulated albedo equal to 0.6 at 1000 nm. Fig. 12 shows that at SSA = 7.4 m² kg⁻¹, $\Delta\alpha_{sim,rough}$ decreases of 0.035 with multiple reflections, which is significant. The impact of multiple reflections is larger for intermediate values of albedo since photons have the same probability to be absorbed or reflected at each collision. Fig. 12 also illustrates that multiple reflections are less sensitive at large SSA, as discussed in Sect. 4.3.1. Hence, it leads to albedo close to 1 and the absorption is too low to trap the photons. Similar results were found in the literature (O'Rawe, 1991; Warren

et al., 1998).

Therefore, for the experiment A-$\eta$27%, we predict that albedo decreases by 0.04 with multiple reflections and by 0.04 with the effective angle effect, i.e. a total albedo decrease of 0.08 due to the presence of surface roughness only. Effective angle effects increase with large $\theta_s$ and low SSA, while the impact of multiple reflections becomes larger when SSA correspond to

intermediate value of albedo in the near-infrared wavelengths. Both effects are stronger when the roughness orientation is perpendicular to the sun.

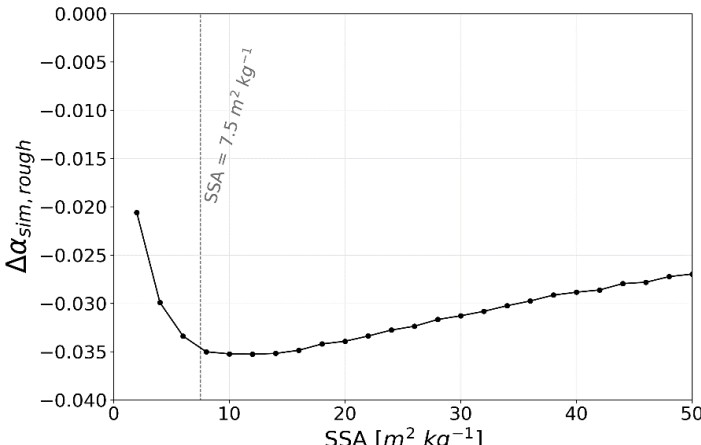

**Figure 12.** $\Delta\alpha_{sim,rough}$ **variations as a function of SSA (m² kg⁻¹). RSRT simulations are computed with the KZ04 configuration at** $\lambda$**=1000 nm, with the initial conditions of the experiment A-$\eta$27% (rectangular shapes, $\theta_s$ = 63°, $\Delta\varphi_r$ =71°, without slope, Table 1).**

**The vertical dotted lines indicate the measured SSA (7.4 m² kg⁻¹).**

### 4.5 Impact on the radiative balance

In this study, the observed albedo change due to the presence of surface roughness may seem low, of the order of a few percent. However, even a small decrease in albedo may strongly impact the radiative balance by increasing the proportion of absorbed energy, estimated with the net short wave radiation ($SW_{net}$). To illustrate the importance of such an albedo decrease

on the radiative balance, we compute $SW_{net}$ using RSRT with the simple approach described in the following. The net short wave radiation in the 0.35 - 4 µm range (in W m⁻²) is estimated with Eq. (17):

$$SW_{net} = \int_{0.3\mu m}^{4\mu m}(1 - \alpha_{dir}(\lambda, \theta_s))Irr_{\text{dir}}(\lambda)d\lambda + \int_{0.3\mu m}^{4\mu m}(1 - \alpha_{diff}(\lambda))Irr_{diff}(\lambda)d\lambda \qquad [17],$$

where $\alpha_{dir}(\lambda, \theta_s)$and $\alpha_{diff}(\lambda)$ are the direct and diffuse albedo, and $Irr_{dir}(\lambda)$ and $Irr_{diff}(\lambda)$ the direct and diffuse solar spectral

irradiance (W m⁻² µm⁻¹) computed with the Santa Barbara DISORT Atmospheric Radiative Transfer (SBDART, Ricchiazzi et

al., 1998). SBDART is an atmospheric model computing radiative transfer within the Earth's atmosphere and at the surface, in clear-sky (direct illumination) and cloudy conditions (diffuse illumination).

The net short wave radiation is estimated with Eq. (17) over the C smooth and C rough 90° surfaces using $\alpha_{sim,\text{smooth}}$ and $\alpha_{sim,\text{rough}}$, respectively, at $\theta_s = 68°$. For this simulation, we assume that there are no impurities in the snow, and that the presence of
roughness is the only cause of the albedo decrease. SBDART is run with a mid-latitude winter atmospheric profile, at 1729 meters high (elevation of the site of experiment C), and at noon.

The broadband albedo simulated by considering surface roughness is 0.05 lower than the one simulated with the smooth surface. It results in an increase of the $SW_{\text{net}}$ from 15 W m$^{-2}$ to 27 W m$^{-2}$ caused by the presence of surface roughness. In other words, the energy absorbed by the snowpack may increase by almost a factor two (+80 %) with the presence of roughness.
Note that this is an illustration of the potential impact of roughness on the $SW_{net}$, more than a real estimate, because RSRT has not been fully validated at wavelength below 600 nm and above 1050 nm, and because we simulate artificial roughness which may not be representative of the whole alpine snowpack. Nevertheless, these results illustrate the necessity to consider surface roughness in the estimation of the surface energy budget. Further work and measurements are needed to validate the radiative balance simulation, and this is out of the scope of this study.
The RSRT model was evaluated with artificial roughness here, and the next step will logically concern natural rough surfaces. An interesting perspective would be to apply this model at a larger scale for remote sensing applications, in particular in complex terrain (mountainous area). Nevertheless, this work will prove challenging since at such a scale, the atmosphere scatterings have to be integrated in the Monte Carlo algorithm which will increase the number of photon hits.

## 6 Summary and perspectives

Four controlled experiments using artificial roughness fields with various geometrical characteristics (fraction of roughness features, orientation, etc.) were studied. Our observations show that the presence of macroscopic surface roughness significantly decrease snow albedo. More specifically:

- Even a low fraction of roughness features ($\eta = 7$ %) causes a detectable albedo decrease up to 0.02 at 1000 nm relative to a smooth surface,
- For higher fractions ($\eta = 27$ % and 63 %), and when the roughness orientation is perpendicular to the sun, the decrease ranges between $0.03 - 0.05$ at 700 nm and of $0.07 - 0.10$ at 1000 nm. The impact is 20% lower when the orientation is parallel to the sun.
- At low SSA (10 m$^2$ kg$^{-1}$), the albedo sensitivity to surface roughness is twice as large at 1000 nm (NIR) than at 700 nm (visible) due to the higher intrinsic absorption of the snow.

We developed a new model to account for surface roughness in snow albedo simulations. RSRT considers both the 3-D geometric effects introduced by roughness and snow optical properties using a Monte Carlo photon transport algorithm. By considering roughness, albedo simulations are improved by a factor 2 compared to those assuming a smooth surface (RMSD of 0.03 at 700 nm and 0.04 at 1000 nm).

Using RSRT, we analysed how the contributions usually affecting albedo interact with the effects of roughness.
Firstly, we investigated the impact of SSA and slope uncertainties in our roughness analysis. The amplitude of roughness effects is insensitive to SSA variations at high SSA. On the contrary, at low SSA, a SSA decrease of 50 % induces the same reduction in albedo that the one due to the presence of roughness. Hence, the albedo decrease due to the presence of roughness is drastically accentuated when SSA is low ($< 10$ m$^2$ kg$^{-1}$) and when the roughness orientation is perpendicular to the sun. This is explained by 1) when the sun elevation is low, the reduction of the local incident angle of roughness sides facing the sun has
more consequences on the albedo when SSA is low (higher absorption of photons), and 2) the impact of multiple reflections is larger when the probability of a photon to be absorbed or reflected is well balanced, which is mainly controlled by a low

SSA in the NIR (albedo ~ 0.6). In addition, the overall slope of the rough surface changes the local incident angle and accentuates roughness effects when the surface aspect is facing away the sun. Therefore, to accurately quantify the effects of roughness, it is necessary to know SSA variations when albedo measurements are acquired and the slope of the surface.

Secondly, the two processes governing roughness effects were quantified separately with RSRT. We showed that the albedo decrease due to the effective angle effect becomes rapidly stronger with $\theta_s$ at large $\theta_s$ ($\theta_s > 50°$) and when $\Delta\varphi_r = 90°$. For instance, the effective angle effect causes a reduction in albedo 40% stronger when $\theta_s$ goes from 63° to 80° for roughness shapes considered here. The impact of multiple reflections is larger for SSA between 8 m$^2$ kg$^{-1}$ and 14 m$^2$ kg$^{-1}$. Thus, the impact of roughness is strongly linked to SSA, slope, the solar zenith angle and the roughness orientation. RSRT provides a useful

tool to better characterize the albedo sensitivity to macroscopic surface roughness.

Roughness effects are significant and many biases are introduced by neglecting these contributions. For approaches considering a smooth surface and using simulated and observed albedo to retrieve SSA, the presence of roughness causes a strong underestimation of SSA, which can be of the order of 20 % for roughness features perpendicular to the sun. Moreover, the albedo decrease leads to an increase of the absorbed energy in the snowpack. In one of our experience, we found that a

decrease of the broadband albedo of 0.05 causes +80 % of additional net short wave radiations relative to a smooth surface. This result highlights the necessity to take into account the roughness effects to compute the surface energy budget. RSRT was evaluated on meter-scale artificial roughness. In further work it will be applied both for natural roughness and at a larger scale in complex terrain (mountainous area).

**Author Contributions:** Fanny Larue, Ghislain Picard, and Laurent Arnaud contributed to the conceptualization and design of the work. All authors contributed to the observation acquisition, analysis, and interpretation of data. Ghislain Picard wrote RSRT, Fanny Larue led the analysis and wrote the manuscript, and all the authors contributed to revisions of the manuscript.

**Aknowledgements**. This project was supported under the EAIIST project, with the financial contribution of the Institut Polaire

Français Paul-Emile Victor (IPEV), the Agence Nationale de la Recherche (ANR programs 1-JS56-005-01 MONISNOW and ANR-16-CE01-0006 EBONI), Equipex CLIMCOR, the Centre de Carottage et de Forage National (C2FN), the Station Alpine Joseph Fourier (SAJF) and the Centre Nationnal d'Etudes Spatiales (CNES). CNRM-CEN and IGE are part of Labex OSUG@2020 (investissement d'avenir – ANR10 LABX56). The authors would like to thank Bertrand Cluzet (Centre d'Études de la Neige, Météo-France) for his help during the field campaigns.


**Data availability:**

The albedo observations and auxiliary data will be assembled in an open datatset to be released on https://persyval-platform.univ-grenoble-alpes.fr after the review process.

**Competing interests.** The authors declare that they have no conflict of interest.

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
