# Peer review of "Snow albedo sensitivity to macroscopic surface roughness using a new ray tracing model"

_The Cryosphere, 2019_

## Referee Comment (RC1) · Anonymous Referee #1 · 20 Nov 2019

Authors measured spectral albedo in a flat smooth and an artificial rough surface, and developed a new ray tracing model to quantify the effects of the macroscopic surface roughness on the snow albedo. Reviewer gives a certain appreciation for the reasons; authors showed that the presence of macroscopic surface roughness significantly decreases snow albedo. Furthermore, snow albedo depends on the fraction of roughness feature, solar zenith angle and relative azimuth angle between the sun and the surface roughness orientation. However, the explanations of some results are insufficient. Particularly, reviewer cannot understand the reason why spectral albedo exceeded 1.0. It is not a realistic in nature. In addition, reviewer is wondering whether the RSRT model can represent the measurement data even in the flat smooth surface from the results of comparison between simulated spectral albedos and measured ones. Thus, it is

questionable whether all simulation including results of sensitivity analyses are true. Reviewer supposes there are new findings about this research (regarding measurement data). Thus, the manuscript would have a merit for the publication in the TC. But, simulation results would be insufficient at this moment. Authors should carefully confirm the results and then provide a detailed explanation or modify the structure of the manuscript.

(Major comments)

1. In Fig. 5, all the simulated spectral albedos exceed 1.0 in the wavelength region of < 700 nm even in the case of the flat smooth surface. Also, the measured spectral albedos exceeded 1.0 in the range of < 870 nm in Fig. 7. These results are not realistic in nature and misleading information. Reviewer recommends explaining the reason why spectral albedos exceed 1.0.

2. Simulated spectral albedos were not consistent with measured ones as a whole. There are some discrepancies between them. For example, the measured variation $Delta\ alpha$ shows a clear dependence on $Delta\ phi\_r$ while the simulated one doesn't (Fig. 8). Reviewer supposes that the measurement values presented here are true. Thus, I am wondering whether the RSRT model provides certain values or not. Authors need to show the agreement between the model and the measurement to present how the proposed model works properly. Otherwise, it could be difficult to achieve the objective of this study which is to quantify the impact of surface roughness on snow albedo.

(General comments)

1. L29: Regarding the sentence "For a typical alpine snowpack ... 27 Wmˆ2).", this estimation was the value at the site C based on the artificial rough surface. Reviewer is wondering if "a typical alpine snowpack" means the natural rough surface in the mountain regions. How does the artificial rough surface represent the natural snow surface in the mountain regions?

2. L40: Snow grain shape is also one of the important factor to control the snow albedo (Tanikawa et al., 2006; Jin et al., 2008) in addition to the physical properties mentioned in the manuscript. Authors should add explanations and cite research papers.

- Jin et al. (2008): Snow optical properties for different particle shapes with application to snow grain size retrieval and MODIS/CERES radiance comparison over Antarctica, Remote Sensing of Environment, 112, 3563-3581.

- Tanikawa et al. (2006): Monte Carlo simulations of spectral albedo for artificial snow-packs composed of spherical and nonspherical particles, Applied Optics, 45, 5310-5319.

3. L199: What does LAP stand for?

4. L201: Reviewer is wondering if measured spectral albedo is relatively high at wavelength range 500–700 nm even in a contaminated snow. This comment might be related to the major one.

5. L202: It would be difficult to say a following sentence "The albedo decrease in the 400-600 nm range is a clear structure of a high LAP concentration". Only small amount of black carbon causes a drastic albedo decrease in the visible regions. Authors should add/modify the explanation properly.

6. L205: Describe the reason why authors chose 700 nm and 1000 nm for the statistical results. The reason is not clear. For example, it would be better to select wavelengths used for satellite remote sensing.

7. L210: How did authors consider the effect of atmosphere in the radiative transfer calculation?

8. L230: In general, the asymmetry factor (g) increased with increasing (decreasing) the snow grain size (SSA) in the near infrared regions. So, g should be linked with the snow grain size (or SSA). This assumption might lead to biases of spectral albedo simulation.

9. L264: It is not clear whether the roughness part (Monte Carlo algorithm) employs the single scattering properties (single scattering albedo, phase function and so on) and/or surface reflectance of snow or not. How does the photon decide "hit" or "not hit"? Random number with snow single scattering albedo or snow reflectance? How does next direction after the scattering (i.e. after the photon hits to the snow grain) decide? Detailed explanations are needed.

10. L463: In Figs. 8a and d, the results $\Delta alpha$ were not symmetry at $\Delta phr\_r$=0. The effect of surface slope caused the asymmetry of $\Delta alpha$ at $\Delta phr\_r$=0? Explanations are needed.

11. L635: This is a rough estimation in a net SW radiation because the validation of the proposed model would not be adequately tested in the visible and shortwave near-infrared region (> 1000 nm). In addition, the effect of snow impurity such as a black carbon and a dust was not considered in the estimation of the net SW radiation. As authors well know, the spectral snow albedo depends on the concentration of snow impurity in the visible region where solar radiation is larger in the relatively cloud free condition. Thus, there would be a large uncertainty in the estimation (there are many parameters to be considered in the estimation, e.g. snow layer (vertical) information). Reviewer supposes that this item is next step.

---

## Referee Comment (RC2) · Anonymous Referee #2 · 15 Jan 2020

SUMMARY Larue and colleagues present both in-situ observations and a Rough Surface Ray Tracer (RSRT) model to assess the impact quantify the impact of surface roughness on snow albedo. Their observations show that surface roughness features have a strong impact at albedo reductions. This impact is already apparent for low roughness values, but becomes more pronounced for higher roughness values, where the albedo reduction depends strongly on the roughness orientation relative to the sun. Besides the observations, Larue and colleagues also introduce for the first time a model that allows to account for surface roughness in snow albedo simulations. Simulations with the model show that albedo simulations are improved by a factor 2 compared to those assuming a smooth surface. The model gives moreover insight in the role of Specific Surface Area (SSA), slope, the solar zenith angle and the roughness orien-

tation. Finally, the paper highlights the necessity to take into account the roughness effects to compute the surface energy budget.

GENERAL COMMENTS The paper of Larue and colleagues touches upon an important topic, is well written, extensively analyzed. As such it build further on earlier work of Warren, Cathles, Pfeffer, Lhermitte and many others, but with the clear novelty that it adds new well designed measurements and the RSRT model that allows to assess the effects in 3D (versus earlier 2D models). Based on these comments I think the paper is well suited and already well written and organised to merit publication in TC. Nevertheless, I have some minor comments that might be addressed in an eventual revised version of the paper.

MINOR COMMENTS L124 "by uniformly pressing a rectangular metal bar into the snow" : What would be the effect of compression and corresponding differences in density/SSA on the observed albedo values. Do you expect this to interfere with the observations? If so/not, why and what would be the effect?

Measured albedo values above 1: the paper shows several figures with spectral albedo values above 1 which is physically impossible. It would be good to explain where these values come from and what it means in terms of uncertainty (also for the rest of the observations and conclusions).

Figure 1: Based on this figure it seems that the sun is oriented North. I know that it is only an illustration and a minor detail, but it might be clearer if the sun is positioned south for norther hemisphere experiments.

Figure 5: Comparison between the simulated smooth and observed albedo values seems to show still some minor contamination by LAP's in shorter wavelengths. Perhaps worthwhile to mention that as well when discussing this graph?

L650 "large scale": it would be good if the authors could already add a discussion point of what the current results would mean for larger scale roughness features and/or how

the conclusions from this paper can (or not) be extrapolated to larger scale roughness features.

---

## Author Comment (AC1) · 12 Mar 2020

The answer to referee 1 is available in PDF format within the Supplement file.

=> Below, the answers to reviewer are written between the '***' symbol.

——-

Anonymous Referee #1

Authors measured spectral albedo in a flat smooth and an artificial rough surface, and developed a new ray tracing model to quantify the effects of the macroscopic surface roughness on the snow albedo. Reviewer gives a certain appreciation for the reasons; authors showed that the presence of macroscopic surface roughness significantly decreases snow albedo. Furthermore, snow albedo depends on the fraction of roughness feature, solar zenith angle and relative azimuth angle between the sun and the surface roughness orientation. However, the explanations of some results are insufficient. Particularly, reviewer cannot understand the reason why spectral albedo exceeded 1.0. It is not a realistic in nature. In addition, reviewer is wondering whether the RSRT model can represent the measurement data even in the flat smooth surface from the results of comparison between simulated spectral albedos and measured ones. Thus, it is questionable whether all simulation including results of sensitivity analyses are true. Reviewer supposes there are new findings about this research (regarding measurement data). Thus, the manuscript would have a merit for the publication in the TC. But, simulation results would be insufficient at this moment. Authors should carefully confirm the results and then provide a detailed explanation or modify the structure of the manuscript.
* * *
First of all, the authors thank the reviewer for the constructive review of the manuscript. In this section, we provide a brief description of the major changes applied in the new version of the manuscript following the reviewer's comments.

In the case of a flat smooth surface, albedo simulations with the RSRT model are the same as simulations using the ART theory (Kokhanovsky and Zege, 2004; see Section 3.1). For dry snow, numerous studies have shown a good agreement between the albedo simulated with the ART theory and observations over smooth surfaces (Dumont et al., 2017; Wang et al., 2017). New sections have been added to clearly evaluate RSRT simulations in presence of surface roughness, and to explain why albedo values may be above 1 in the visible range. Overall, a strong effort has been made to make the text clearer (see the track change version of the manuscript). As suggested by the reviewer, we changed the structure of the manuscript by adding two new sections (new 2.4 and 4.1) to explain results with more details.

[Figure]

New Section 4.1:

An entire section has been introduced at the beginning of the Results section to investigate the performance of the RSRT model, before the sensitivity analysis. We think that the accuracy of simulations are more transparent now. In particular, we explain why the simulated spectra do not overlap perfectly observed spectra. - In the visible, measurements and simulations differ in the 600-700 nm range probably because of the concentration of impurities which are not considered in our simulations (since not measured). Note that the RSRT model is capable of accounting for impurities if they are measured. - In the NIR domain it is probably because of a small bias in SSA measurements (10% uncertainty).The albedo-SSA relation in the NIR is linear, meaning that a variation of 10% of SSA induces a variation of +- 0.01 of albedo. Here, the difference between simulated and measured spectra in the NIR domain is below 0.01 and may come from SSA uncertainties. The impact of measurement errors in our sensitivity analysis is discussed in detail in Section 4.3. Hence, measured and simulated spectra differ slightly due to inherent measurement errors. Nevertheless, the RSRT model improves the spectral albedo simulations by taking roughness into account, compared to simulations which neglect them (i.e. considering a flat surface, see Figures 6 and 7). Considering all observations, albedo simulations with the RSRT model are improved by a factor 2 by accounting for surface roughness compared to those neglecting them, which is significant.

To the best of our knowledge, this is the first model capable of simulating spectral albedo taking into account the actual surface roughness, the slope and snow optical properties using a Monte Carlo photon transport algorithm.

New Section 2.1Âă:

We detailed why the measured and simulated albedo values may exceed 1 in the visible range with a new Section in the Methodology section. Explanations are given further in the Major Comments section.
* * *
Major comments

1. In Fig. 5, all the simulated spectral albedos exceed 1.0 in the wavelength region of < 700 nm even in the case of the flat smooth surface. Also, the measured spectral albedos exceeded 1.0 in the range of < 870 nm in Fig. 7. These results are not realistic in nature and misleading information. Reviewer recommends explaining the reason why spectral albedos exceed 1.0.
* * *
The reason why we have albedo values over 1 is the presence of a sun-facing slope, and because here we consider apparent albedo, and not intrinsic albedo (i.e. albedo of a flat terrain): When the terrain is not flat, the horizontal sensor acquiring the snow reflectance is not perfectly parallel to the snow surface, and thus the ratio of the readings from the sensor when measuring the incoming irradiance and the snow reflectance (called the apparent or measured albedo) is different from the intrinsic surface albedo (true albedo = with a perfectly flat surface). Picard et al. (2020) fully detailed these slope effects on the measured albedo, which may be over 1 when the slope is facing the sun, even with perfect instruments and even for weak slopes ∼2°. This slope effect, inducing measured spectral albedo in the visible range above 1, has been observed in numerous previous studies (Grenfell et al, 1994Âă; Wutttke et al., 2006Âă; Dumont et al., 2017), and it has been demonstrated that it is because there is a higher interception probability of the sun beam by these slopes facing the sun compared to horizontal surfaces (Picard et al., 2020). Of course, the apparent albedo does not represent the well-known reflectance, the energy is not conserved and it must not be used for energy budget calculations (where a flat terrain has to be considered). It is correct to simulate the apparent albedo here since the goal was to validate RSRT with apparent albedo observations.

To show an example of the slope effect on measurements, Figure 1 illustrates a comparison of two measured spectra acquired over two smooth surfaces in the French Alps in clear sky condition: one with a slope facing away the sun and one with a sun-facing slope. Snow conditions at the surface were similar, with close SSA values. The presence of the slope facing the sun induces a distortion of the spectra, with values above 1 in the visible and a concave spectral shape. When a small slope is facing back the sun, the pattern of the spectra shows no distortions, and the presence of a slope is difficult to detect. If they are not taken into account, slopes may induce strong biases in snow parameters estimated from optical measurements.

SEE FIGURE 1

Legend of Figure 1: Measured spectral albedo with Solalb over two smooth titled surfaces having similar snow properties. Measurements are acquired in clear sky conditions. One surface has a $5°$ slope facing the sun (blue line) and the other has a $3°$ slope facing away the sun (red line).

Several changes have been made in the text to be clearer on this point, and we detailed them in the following.

- The notion of observed apparent albedo is introduced in Section 2.2

L. 180Âă: "ÂăThe observed apparent albedo, hereinafter referred to as $\alpha$obs, is the processed spectrum measured with Solalb, considering the sensor in a horizontal position (Sicart et al., 2001)."

- In Section 2.3 we cancelled the part explaining slope measurements, and we added an entire new section further (now Section 2.4) to explain more clearly the impact of slope on the measured albedo, and slope measurements. The following text has been introduced, with a new figure (now Figure 4) to illustrate our arguments:

Line 212: " In the case of a tilted surface, Solalb is not perfectly parallel to the snow surface, and therefore the ratio of values acquired by the sensor when it measures the downwelling and the upwelling spectral irradiance (i.e $\alpha$ obs ) differs from the intrinsic

surface albedo (called true albedo in previous studies, i.e measured with a perfectly flat surface) (Picard et al., 2020). Indeed, when the sensor is horizontal, the titled surface receives sun radiation with a different incidence angle and is viewed with a reduced solid angle by the sensor (Grenfell et al, 1994; Wutttke et al., 2006; Dumont et al., 2017). With surfaces having a sun-facing slope, it has been demonstrated that measured albedo values may be over 1 in the visible range (the spectra are distorted with a concave shape), because there is a higher interception probability of the sun beam by these slopes facing the sun compared to horizontal surfaces (Picard et al., 2020). Therefore, measured albedo may exceed 1 in the visible range, while it is unrealistic for the intrinsic albedo that is strictly ranged between 0 and 1. Of course, the measured albedo does not represent the well-known reflectance, the energy is not conserved and it must not be used for energy budget calculations (where a flat terrain has to be considered). In this study, surfaces of experiments A, B and C have small sun-facing slopes (Table 3), and the slope effects does not have to be neglected in albedo simulations since even a small slope (Dumont et al 2017)"

- Concerning field experiments, there were no perfect flat surfaces and even the studied smooth surfaces had small slopes (it is very difficult to find a perfectly flat surface in the field). In experiments A, B, C (Fig. 5 and 7) measurements are above 1 in the visible range because surfaces have a sun-facing slope.

L.149Âă: "There are no perfectly flat surfaces in this study since it is difficult to find such surfaces in the field, and thus all studied surfaces have small slopes. In particular, it is noteworthly that experiments A, B and C have a small sun-facing slope."

- Concerning simulations, we modelled the apparent/measured albedo in order to be compared with measurements. The distorsion of the spectra due to the presence of the slope is modeled with the K factor introduced by Dumont et al. 2017.

L. 272 "ÂăAs shown by Dumont et al. (2017), the K factor is the relative change in the cosine of the sun effective incident angle to the slope, and makes it possible to

reproduce the distortion of the spectra due to the presence of the slope (with potential albedo values above 1 in the case of a sun-facing slope)."

- In the Results section, we added the following sentence to explain why the albedo values are above 1 when lambda < 700nmÂă:

For experiments A and B, in Section 4.1, L. 419: "Both surfaces have a sun-facing slope (3.1° for experiment A and 3.6° for experiment B, see Table 1), so albedo values above 1 in the visible range are not surprising (see Sect. 2.4). "

For experiment C en Section 4.1Âă: L.451: "For experiment C, apparent albedo exceeds 1 in the visible range because of the presence of a sun-facing slope (3.3° - 4°, see Table 1)."
* * *
2. Simulated spectral albedos were not consistent with measured ones as a whole. There are some discrepancies between them. For example, the measured variation $Delta alpha$ shows a clear dependence on $Delta phi\_r$ while the simulated one doesn't (Fig. 8). Reviewer supposes that the measurement values presented here are true. Thus, I am wondering whether the RSRT model provides certain values or not. Authors need to show the agreement between the model and the measurement to present how the proposed model works properly. Otherwise, it could be difficult to achieve the objective of this study which is to quantify the impact of surface roughness on snow albedo.
* * *
- Agreements between the model and the measurements are shown in the new section 4.1 'RSRT evaluation'. We show that simulations accounting for the surface roughness are more accurate by about a factor 2 at 700 nm and 1000 nm compared to those neglecting them (i.e ART theory with a flat terrain), which is significant.

L.460: "ÂăConsidering all observations, albedo simulations with the RSRT model are

improved by a factor 2 by accounting for surface roughness ($\alpha$sim,rough) at 700 nm and 1000 nm compared to those neglecting them ($\alpha$sim,smooth), with an average RMSD of 0.03 at 700 nm and 0.04 at 1000 nm (Table 3). To the best of our knowledge, this is the first model capable of simulating spectral albedo taking into account the actual surface roughness, the topography and snow optical properties using a Monte Carlo photon transport algorithm."

In addition, we modified Figures 6 and 7 to illustrate the spectral performance of the RSRT model for all experiments. The differences between measured and simulated albedo spectra are explained as followsÂă:

L. 426: "For both experiments, the pattern of the measured spectra between 600-700 nm is probably led by the presence of impurities (not visible to the naked eye on the field). Previous studies showed that a even a small concentration of snow LAPs induces a drastic decrease of the albedo in the visible range (Warren, 1984; Dumont et al., 2017), and may explain why measurements and simulations differs in the 600-700 nm range. Moreover, the two spectra do not overlap perfectly in the NIR domain, but differences are below 0.01, and it is probably because of a small bias on SSA measurements (10% uncertainty). Overall, taking into account the measurement errors, $\alpha$sim,rough spectra reproduces the observed spectra well for both experiments and the RSRT model improves the spectral albedo simulations by accounting for roughness, compared to those which neglect them (Fig. 6)."

- To answer to the reviewer about the Fig. 8 (now Fig. 9)Âă: there is a dependence between measured variation $Delta alpha$ and $Delta phi\_r$, but it is misleading due to the presence of different contributions that change the albedo. It looks like measured $Delta alpha$ increases when $Delta phi\_r$ goes from 42° to 72° (i.e. becomes closer to 90°). It means that the roughness effect on albedo values is lower when the roughness orientation is closer to be perpendicular to the sun than parallel. This measured trend is opposite of what we find in the litterature, as shown by Warren (1998). The Figure 2 (from Warren, 1998) illustrates the reduction in albedo when the roughness

orientation becomes closer to 90°. RSRT simulations reproduce well this trend in Figure 9c and 9d. Authors guess that the measured trend is due to the melting observed in the field, resulting in a smoothing of roughness features thoughought the day. Hence, we can not conclude on this observed trend since several contributions disturbed the measured albedo.ĂăWe detailed it in the text as followsĂă:

L. 556: "ĂăIn Fig. 9c and 9f, the $\Delta\alpha$obs increases when $\Delta\varphi$r goes from 42° to 72°, while in theory it should decrease when $\Delta\varphi$r approaches 90°. A possible explanation is that melting was observed at the surface in the field, resulting in a smoothing of our roughness shapes during the day, which attenuates the roughness effect on albedo values. Therefore, we can not conclude on this observed trend since several contributions disturbed the measured albedo.Ăă"

see FIGURE 2

Legend of Figure 2: Figure 13 in Warren et al (1998). Effect of sastrugi on albedo, from the Monte-Carlo radiative transfer modeling of O'Rawe [1991]. Plotted is the change in albedo as a function of the height-to-width ratio of rectangular sastrugi with spacing equal to width. Simulations are performed with an illumination by a direct beam at 60° from the nadir, for four different sastrugi azimuths; flat-surface albedo = 0.8.
* * *
General comments

1. L29: Regarding the sentence "For a typical alpine snowpack ... 27 WmËĘ2).", this estimation was the value at the site C based on the artificial rough surface. Reviewer is wondering if "a typical alpine snowpack" means the natural rough surface in the mountain regions. How does the artificial rough surface represent the natural snow surface in the mountain regions?
* * *
Even if the size of the artificial roughness features used in this study are not exaggerated compared to what can be observed in the French Alps, natural patterns in the field are strongly correlated to the wind, sun exposition, and topography amongst others. The surface roughness pattern has high spatial variability in mountainous areas and is difficult to quantify in the field. Further studies (including in situ measurements) are needed to determine what is representative for a natural snow surface in Alpine areas, using photogrammetric measurements for instance, but this is out of the scope of the present study.

To be clearer in the abstract we changed the sentence as followsÂă:

Line 28Âă: "For a snowpack where we artificially created surface roughness, we showed that a broadband albedo decrease of 0.05 may cause an increase of the net short wave radiation of 80 % (from 15 W m-2 to 27 W m-2). "

\*\*\*

2. L40: Snow grain shape is also one of the important factor to control the snow albedo (Tanikawa et al., 2006; Jin et al., 2008) in addition to the physical properties mentioned in the manuscript. Authors should add explanations and cite research papers.

- Jin et al. (2008): Snow optical properties for different particle shapes with application to snow grain size retrieval and MODIS/CERES radiance comparison over Antarctica, Remote Sensing of Environment, 112, 3563-3581.

- Tanikawa et al. (2006): Monte Carlo simulations of spectral albedo for artificial snowpacks composed of spherical and nonspherical particles, Applied Optics, 45, 5310-5319.

\*\*\*

It is true that the impact of snow grain shape on snow albedo can be significant, and this question has been adressed by the team through several publications (Picard et al., 2009Âă; Libois et al., 2013, 2014). We added this factor line 41, as follows (Line 40)Âă:

"Âǎ Snow spectral albedo generally depends in a complex way on several factors, including 1) the snow physical and chemical properties, mainly the Specific Surface Area of snow grains (SSA, Gallet et al., 2009), the snow grain shapes (Tanikawa et al., 2006; Jin et al., 2008; Libois et al., 2013, 2014) and the concentration of snow Light Absorbing Particles (referred to as LAP, Skiles et al., 2018)"

And we added the research papers in the reference section.

\*\*\*

3. L199: What does LAP stand for?

\*\*\*

The LAP acronym was described in the introduction Line 43Âǎ: "the concentration of snow Light Absorbing Particles (called LAP, Skiles et al., 2018)"

LAP describes several types of impurities such as mineral dust, black carbon or algae. To be clearer, we changed the sentence Line 197 as followsÂǎ:

" the concentration of Light Absorbing Particles (called LAP), such as mineral dust and black carbon, was not measured although they strongly lower the spectral signature in the visible range (Warren, 1982) "

\*\*\*

4. L201: Reviewer is wondering if measured spectral albedo is relatively high at wave-length range 500–700 nm even in a contaminated snow. This comment might be related to the major one.

\*\*\*

The high measured albedo in the visible is due to the presence of a slope facing the sun. This point is explained in detail in the point 1) of the Major Comments section.

\*\*\*

5. L202: It would be difficult to say a following sentence "The albedo decrease in the 400-600 nm range is a clear structure of a high LAP concentration". Only small amount of black carbon causes a drastic albedo decrease in the visible regions. Authors should add/modify the explanation properly.
* * *
It is true that the presence of each LAP type (black carbon, mineral dust, etc.) causes a drastic albedo decrease. To be clearer, we modified the sentence line 201 as followsĂă:

"The albedo decrease in the 400-600 nm range is a clear signature of the presence of snow impurities. Even a small amount of LAP led to a high decrease of the albedo in the visible domain (Tuzet et al 2019).."
* * *
6. L205: Describe the reason why authors chose 700 nm and 1000 nm for the statistical results. The reason is not clear. For example, it would be better to select wavelengths used for satellite remote sensing.
* * *
Future work will address the application of the model over large-scale natural surfaces including the atmosphere, but for now the validity of this model for satellite remote sensing applications is out of the scope of this paper.

The selection of the two wavelengths (700nm and 1000nm) is explained by adding the following sentences:

Line 371: "The main goal of this study is to quantify the roughness effect on albedo values and to determine if this effect is frequency dependent. Therefore, statistical results are given at two frequencies: one in the visible domain at 700 nm and one in the NIR domain at 1000 nm. These wavelengths were chosen to be as representative as possible of each domain for the sensitivity analysis. The relation between roughness effect

and SSA is investigated at 1000 nm since at this wavelength the albedo sensitivity to SSA is larger (Dominé et al., 2006)."

700 nm was chosen randomly but values are relatively stable in the 600-700 nm range.

\*\*\*

7. L210: How did authors consider the effect of atmosphere in the radiative transfer calculation?

\*\*\*

This is explained in the 'simulation framework' section (Sect 3.3). The effects of the atmosphere are not taken into account in the Monte Carlo algorithm. The only atmospheric parameter used in the model is the diffuse-to-totalÂăillumination ratio (which depends on atmospheric conditions) to compute the apparent albedo by combining the direct and diffuse albedo components. This parameter was measured in the field shortly after the albedo measurement by screening the sun to record the diffuse irradiance, the total irradiance being measured with the sensor looking upward (see Section 2.2, Line 179).

We added the following sentence after the equation 5, Line 265: "Âăwhere rdiff-tot($\lambda$, $\theta$s) is the ratio of diffuse-to-total illumination at wavelength $\lambda$ and at $\theta$s, measured in the field shortly after each albedo measurements.Âă"

In order to be clearer in the 'simulation framework' section, we modified the sentenceÂă:

Line 353: "RSRT outputs the snow spectral albedo, either in direct or diffuse illumination conditions: $\alpha$dir,rough($\lambda$, $\theta$s) and $\alpha$diff,rough($\lambda$), respectively, considering that the plane of the mesh is perfectly flat. Then, $\alpha$dir,rough($\lambda$, $\theta$s) and $\alpha$diff,rough($\lambda$) are combined with Eqs. (6) and (7) to simulate the apparent snow albedo of a titled rough surface, called $\alpha$sim,rough($\lambda$, $\theta$s), and therefore the simulated apparent albedo accounts for the slope characteristics and the surface roughness. Each simulation assumes

clear sky conditions, and no atmosphere is considered in the Monte Carlo algorithm. The only atmospheric parameter used in the model is the diffuse-to-totalÂăillumination ratio (which depends on atmospheric conditions). This parameter was measured in the field at each albedo acquisition (see Sect. 2.2). At our scale, the effect of the atmosphere is negligible between the sensor and the surface. Future work will focus on setting up the atmosphere in RSRT for applications over large-scale natural surfaces (mountainous areas)."
* * *
8. L230: In general, the asymmetry factor (g) increased with increasing (decreasing) the snow grain size (SSA) in the near infrared regions. So, g should be linked with the snow grain size (or SSA). This assumption might lead to biases of spectral albedo simulation.
* * *
It is true that g should be directly linked to the SSA, but the asymetry factor is difficult to measure in the field. In the RSRT model we use B (the absorption enhancement parameter) and g (the asymetric factor) to describe the snow grain shape and these parameters are assumed to be constant (i.e. a single homogeneous layer). Neverthless, we used values adapted for an Alpine snowpack and estimated by Libois et al 2014 as followsÂă:

By combining simulations and measurements of reflectance and irradiance (and not visual observation of snow grains) on an extensive set of snow samples taken in the laboratory and in the fieldÂă (French Alps and Antarctica), they experimentally found a B value adapted to describe an 'optical grain size'. Then using the correlation between B and 1-g (see Fig. 1 Libois et al. 2014), they deduced g. Thus, they have shown that using B=1.6 and g=0.86 to model snow optical properties is more realistic rather than considering spherical grains as often done.

To be clearer, we added the following sentence Line 258:

" B and g are the snow shape coefficients and are assumed to be constant (i.e. the snowpack is a single homogeneous layer). Theoretically, g should be directly linked with the SSA, but as g is difficult to measure in the field, we used values estimated by Libois et al. (2014), which combined simulations and in situ measurements of reflectance in Antarctica and the French Alps. They found that using B = 1.6 and g = 0.86 is more realistic to model snow optical properties rather than considering spherical grains as often done."

Picard et al. 2009 have shown that the uncertainty on SSA measured with reflectance is about 20 $\mathring{A}$ ă% if the snow grain shape is not known. But this value was over-estimated since calculated using two extrem theoretical shapes (spheres/cubics) that are not found in natural snow (which is more like a mixture). In the present study, we assumed that the error on measured SSA to be about 10% (as estimated by Arnaud et al. 2011), and the analysis of the impact of SSA uncertainties on our roughness effect is discussed in Section 4.3.1.

\*\*\*

9. L264: It is not clear whether the roughness part (Monte Carlo algorithm) employs the single scattering properties (single scattering albedo, phase function and so on) and/or surface reflectance of snow or not. How does the photon decide "hit" or "not hit"? Random number with snow single scattering albedo or snow reflectance? How does next direction after the scattering (i.e. after the photon hits to the snow grain) decide? Detailed explanations are needed.

\*\*\*

- To decide if the photon is absorbed or reflected, two configurations are available (KZ04 and Lambertian). The KZ04 configuration employs the single scattering properties while the Lambertian configuration uses a constant surface reflectance of snow

TCD

(i.e. an ideal diffusionÂă: albedo is the same, whatever the incidence angle). We detailed it in Step 2 (line 310), but to be clearer we introduced this notion earlier by adding the following sentenceÂă:

Line 291: "Photons are either absorbed or reflected at each hit according to the facet albedo value (Iwabuchi, 2006), that is estimated with the single scattering properties in case of the KZ04 configuration, or as a constant snow reflectance in case of the Lambertian configuration. "

- How does the photon decide "hit" or "not hit"?

In step 1 we detailed the process of 'hit' or 'not hit' by adding the following sentenceÂă:

Line 297: "Basically, it uses a simple recursive intersection routine to test if the photon hits or does not hit the bounding volume, and when positive, the list of triangles is tested (Wald et al., 2007)."

We added the following the reference: Ingo Wald, Solomon Boulos, and Peter Shirley. 2007. Ray tracing deformable scenes using dynamic bounding volume hierarchies. ACM Trans. Graph. 26, 1 (January 2007), 6–es. DOI:https://doi.org/10.1145/1189762.1206075

- How does next direction after the scattering (i.e. after the photon hits to the snow grain) decide?

Each facet is treated as a snow surface, and the next direction is computed according to the BRDF distribution, depending of the incident angle and snow properties, whereby the next direction is sensitive to the asymmetry of the scattering. The scattering within a few degrees of the forward direction is much more probable than scattering to other angles (Warren, 1982).

The BRDF computation is detailed in Step 3, and we added explanations with the following sentenceÂă:

Line 310: "thus, the next direction after the scattering depends of the incident angle of the photon and snow properties. It is sensitive to the asymmetry of the scattering, and the scattering within few degrees of the forward direction is much more probable than scattering to other angles (Warren, 1982)."
* * *
10. L463: In Figs. 8a and d, the results $Delta alpha$ were not symmetry at $Delta-phr\_r$=0. The effect of surface slope caused the asymmetry of $Delta alpha$ at $Delta-phr\_r$=0? Explanations are needed.
* * *
The effect of surface slope in our sensitivity analysis is reduced by taking a smooth surface with a similar slope to that of the rough surface, so by computing rough-smooth albedo we canceled slope effects. In this case (experiment C), the asymmetry is more an albedo insensitivity to small variations of roughness orientation and it is explained by high SSA values (i.e. lower absorptions). The SSA impact is fully detailed in Section 4.3.1.

Line 539: "ÂăHowever, for the C rough 90° experiment (Fig. 8b and 8e), $\Delta\varphi$r varies from 50° to 122° and $\Delta\alpha$obs does not show a strongest albedo reduction around 90°. Similarly, for C rough 0° (Fig. 8a, and 8d), $\Delta\alpha$obs were not symmetrical to $\Delta\varphi$r = 0°. This is caused by two contributions that overlap the roughness effects: the slope and SSA values. Here we selected a smooth surface with a similar slope to that of the rough surface, so as to minimize the impact of this contribution by comparing rough-smooth albedo ($\Delta\alpha$obs). The slope sensitivity to roughness effects is studied in Section 4.3.2. The SSA is particularly high for this experiment ($\sim$100 m2 kg-1). It induces lower absorptions (Warren et al., 1998), and may explain the albedo insensitivity to small variations of roughness orientation. Instead of a clear dependence between $\Delta\alpha$obs and $\Delta\varphi$r, $\Delta\alpha$obs pattern shows oscillations, probably caused by the weak differences in snow properties between the smooth and the rough surfaces. Indeed, SSA was

measured over the smooth surface to be representative, while SSA values over rough surfaces evolved with spatial variations in the concavities according to the received illumination. The SSA sensitivity to roughness effects on albedo measurements is investigated in Section 4.3.1. [. . .] "
* * *
11. L635: This is a rough estimation in a net SW radiation because the validation of the proposed model would not be adequately tested in the visible and shortwave near-infrared region (> 1000 nm). In addition, the effect of snow impurity such as a black carbon and a dust was not considered in the estimation of the net SW radiation. As authors well know, the spectral snow albedo depends on the concentration of snow impurity in the visible region where solar radiation is larger in the relatively cloud free condition. Thus, there would be a large uncertainty in the estimation (there are many parameters to be considered in the estimation, e.g. snow layer (vertical) information). Reviewer supposes that this item is next step.
* * *
=> This part is a discussion of the potential albedo impact on the radiative balance. Authors assume that this is a rough estimation in a net SW radiation, but there is a strong interest to have an order of magnitude of the roughness effect on the absorbed energy.

To be clearer, we modified some sentences, and added some explanationsÂă:

Line 29 in the abstractÂă: "ÂăFor a snowpack where we artificially created surface roughness, we showed that a broadband albedo decrease of 0.05 may cause an increase of the net short wave radiation of 80 % (from 15 W m-2 to 27 W m-2)."

Line 241: "ÂăA simple approach is applied to illustrate the impact of roughness on the quantity of energy absorbed in the snowpack (Sect. 3.4)"

Line 740: "The broadband albedo simulated by considering surface roughness is 0.05

lower than the one simulated with the smooth surface. It results to an increase of the SWnet from 15 W m-2 to 27 W m-2 caused by the presence of surface roughness. In other words, the energy absorbed by the snowpack may increase by almost a factor two (+80 %) with the presence of roughness. Note that this is an illustration of the potential impact of roughness on the SWnet, more than a real estimate, because RSRT has not been fully validated at wavelength below 600 nm and above 1050 nm, and because we simulate artificial roughness which may not be representative of the whole alpine snowpack. Nevertheless, these results illustrate the necessity to consider surface roughness in the estimation of the surface energy budget. Further work and measurements are needed to validate the radiative balance simulation, and this is out of the scope of this study." ***

Please also note the supplement to this comment:
https://www.the-cryosphere-discuss.net/tc-2019-179/tc-2019-179-AC1-supplement.pdf

———————————————————

[Figure]

**Fig. 1.**

[Figure]

**Fig. 2.**

---

## Author Comment (AC2) · 12 Mar 2020

The answer to referee 2 is available in PDF format within the Supplement file.

=> Below, the answers to reviewer are written between the '***' symbol.

——-

Anonymous Referee #2

SUMMARY

Larue and colleagues present both in-situ observations and a Rough Surface Ray Tracer (RSRT) model to assess the impact quantify the impact of surface roughness on snow albedo. Their observations show that surface roughness features have a strong

impact at albedo reductions. This impact is already apparent for low roughness values, but becomes more pronounced for higher roughness values, where the albedo reduction depends strongly on the roughness orientation relative to the sun. Besides the observations, Larue and colleagues also introduce for the first time a model that allows to account for surface roughness in snow albedo simulations. Simulations with the model show that albedo simulations are improved by a factor 2 compared to those assuming a smooth surface. The model gives moreover insight in the role of Specific Surface Area (SSA), slope, the solar zenith angle and the roughness orientation. Finally, the paper highlights the necessity to take into account the roughness effects to compute the surface energy budget.

GENERAL COMMENTS

The paper of Larue and colleagues touches upon an important topic, is well written, extensively analyzed. As such it build further on earlier work of Warren, Cathles, Pfeffer, Lhermitte and many others, but with the clear novelty that it adds new well designed measurements and the RSRT model that allows to assess the effects in 3D (versus earlier 2D models). Based on these comments I think the paper is well suited and already well written and organised to merit publication in TC. Nevertheless, I have some minor comments that might be addressed in an eventual revised version of the paper.

MINOR COMMENTS

R1- L124 "by uniformly pressing a rectangular metal bar into the snow" : What would be the effect of compression and corresponding differences in density/SSA on the observed albedo values. Do you expect this to interfere with the observations? If so/not, why and what would be the effect?

\*\*\*

By compacting the snow, we locally increase the snow density at the surface and it may lead to a small decrease of the SSA (Legagneux et al., 2002Ăă; Dominé et al., 2007..).

It is true that for experiments A and B, we may have disturbed surface SSA observations in the cavities by pressing the bar into the snow. But as the compaction was weak (2cm), and the SSA values were small (7.2 and 4.5 $m^2$/kg before the compaction), we can consider that the observed albedo values were not, or weakly, affected by the compaction.

To be clearer, we added the following sentence line 190:

"Note that compacting to create the roughness features may have lowered the SSA locally. As the compaction was small (2 cm depth), and as the SSAs were initially low over the studied surfaces, we assumed here that the effect of the compaction on the observed albedo is negligible."

For experiment C (fresh snow), we measured 3 surface SSAs at each albedo acquisition to have a good representativeness: over the sides facing, and facing away from the sun, and over the smooth surface (between the cavities). The differences of SSA measured over each surface are lower than 10Âǎ%. We took the mean SSA to compute the albedo, as for each studied area the percentage of surfaces facing, facing away from the sun and smooth are similar. To be clearer on this point we added the following sentence in the paperÂǎ(line 192)Âǎ:

"For the two experiments C and D, three SSA values were measured at the surface at each albedo acquisition: two in the cavities (one over the side facing the sun, one over the side facing away the sun) and one over the smooth surface between cavities. The standard deviation of these three SSA is always lower than 10% of the mean SSA, showing that the compaction effect is negligible compared to measurements uncertainties. The mean of these three SSA values is used in our albedo simulations."
* * *
R2- Measured albedo values above 1: the paper shows several figures with spectral albedo values above 1 which is physically impossible. It would be good to explain

where these values come from and what it means in terms of uncertainty (also for the rest of the observations and conclusions).

\*\*\*

See comment 1 of Reviewer 1. We introduced a new section (section 2.4) and a new figure (new Figure 4) to fully explain why albedo values may exceed 1 in the visible range. It is because our studied surfaces presented small slopes, and we measured the apparent albedo (with a sensor placed horizontally, over a titled terrain), whereby it differs from the true albedo (strictly ranges between 0-1, with a perfectly flat surface).

\*\*\*

R3- Figure 1: Based on this figure it seems that the sun is oriented North. I know that it is only an illustration and a minor detail, but it might be clearer if the sun is positioned south for norther hemisphere experiments.

\*\*\*

We modified the Figure 1 to position the sun South in the illustration (see new Fig. 1).

\*\*\*

R4- Figure 5: Comparison between the simulated smooth and observed albedo values seems to show still some minor contamination by LAP's in shorter wavelengths. Perhaps worthwhile to mention that as well when discussing this graph?

\*\*\*

This is true, we changed the sentence line 426 to mention the weak contamination by LAP's in shorter wavelengthsÂǎ: "For both experiments, the pattern of the measured spectra between 600-700 nm is probably led by the presence of impurities (not visible to the naked eye on the field). Previous studies showed that the even a small concentration of snow LAPs induces a drastic decrease of the albedo in the visible range (Warren, 1984; Dumont et al., 2017), and may explain why measurements and

simulations differs in the 600-700 nm range. "
* * *
R5- L650 "large scale": it would be good if the authors could already add a discussion point of what the current results would mean for larger scale roughness features and/or how the conclusions from this paper can (or not) be extrapolated to larger scale roughness features.
* * *
The RSRT model can be used at a larger scale if driven by an adapted DEM. Nevertheless, at this large scale, the most challenging work would be to include the atmospheric effects in the Monte Carlo algorithm. However this would strongly increase the computing time by increasing the number of photon hits drastically. The algorithm needs to be well optimized.

To be clearer on this point, in the simulation framework section we explain that the atmosphere is not directly taken into account in the Monte Carlo algorithm, by adding the following sentence Line 356:

"Each simulation assumes clear sky conditions, and no atmosphere is considered in the Monte Carlo algorithm. The only atmospheric parameter used in the model is the diffuse-to-total illumination ratio (which depends on atmospheric conditions). This parameter was measure in the field at each albedo acquisition. At our small scale, the effect of the atmosphere is negligible between the sensor and the surface. Future work will focus on setting up the atmosphere in RSRT for applications over large-scale natural surfaces (mountainous areas)."

To discuss the future work concerning the model adaptation at larger scale, we added the following sentences at the end of the section 4.5, line 748:

"The RSRT model was evaluated with artificial roughness here, and the next step will logically concern natural rough surfaces. An interesting perspective would be to apply
this model at a larger scale for remote sensing applications, in particular in complex terrain (mountainous area). Nevertheless, this work will prove challenging since at such a scale, the atmosphere scatterings have to be integrated in the Monte Carlo algorithm which will drastically increase the number of photon hits (i.e the computing time)."

\*\*\*

Please also note the supplement to this comment:
https://www.the-cryosphere-discuss.net/tc-2019-179/tc-2019-179-AC2-supplement.pdf

---

## Author Comment (AC3) · 12 Mar 2020

Dear Editor-in-chief

We would like to thank the reviewers for their detailed comments which were useful in producing an improved manuscript. Below, we carefully responded to all the questions and comments.

A strong effort has been made to facilitate a deeper understanding of this paper (methodology better described, more explanations given in the Results section. . .). We think that this new version is significantly improved compared the previous submission.

Please, do not hesitate to contact me if you have any further questions or comments.

[Figure]

Regards,

Fanny Larue

---

## Author Response (AR1)

**Letter to the editor**

**Revision of manuscript number (doi)**: 10.5194/tc-2019-179

Dear Editor-in-chief,

Please find attached to this letter the revised version of the manuscript. We would like to thank the reviewers for their detailed comments which were useful in producing an improved version of the manuscript. Below, we carefully responded to all the questions and comments.

A strong effort has been made to facilitate a deeper understanding of this paper (methodology better described, more explanations given in the Results section…). We think that this new version is significantly improved compared the previous submission.

In particular, in the revised manuscript we introduced an entire new section (new Section 2.4) to carefully detail the distinction between apparent albedo and intrinsic albedo. In this section we explain that we simulate and measured the apparent albedo of titled surfaces, instead of the intrinsic albedo (ie flat surfaces), and as our experiments have small sun-facing slopes this can lead to albedo greater than 1 in the visible range.

The main changes can be tracked on the track-change document below (after the point-by-point response to the reviews).

Please, do not hesitate to contact me if you have any further questions or comments, or if anything is missing from the submission of the revised manuscript.

Regards,

Fanny Larue
* * *
*-- Comments and questions of reviewers in black*
*-- Answer to reviewers in blue*

Authors measured spectral albedo in a flat smooth and an artificial rough surface, and developed a new ray tracing model to quantify the effects of the macroscopic surface roughness on the snow albedo. Reviewer gives a certain appreciation for the reasons; authors showed that the presence of macroscopic surface roughness significantly decreases snow albedo. Furthermore, snow albedo depends on the fraction of roughness feature, solar zenith angle and relative azimuth angle between the sun and the surface roughness orientation. However, **the explanations of some results are insufficient**. Particularly, reviewer cannot understand the reason why spectral albedo exceeded 1.0. It is not a realistic in nature. In addition, **reviewer is wondering whether the RSRT model can represent the measurement data even in the flat smooth surface** from the results of comparison between simulated spectral albedos and measured ones. Thus, it is questionable whether all simulation including results of sensitivity analyses are true. Reviewer supposes there are new findings about this research (regarding measurement data). Thus, the manuscript would have a merit for the publication in the TC. But, simulation results would be insufficient at this moment. **Authors should carefully confirm the results and then provide a detailed explanation or modify the structure of the manuscript.**

First of all, the authors thank  the reviewer for the constructive review of the manuscript. In this section, we provide a brief description of the major changes applied in the new version of the manuscript following the reviewer's comments.

In the case of a flat smooth surface, albedo simulations with the RSRT model are the same as simulations using the ART theory (Kokhanovsky and Zege, 2004; see Section 3.1). For dry snow, numerous studies have shown a good agreement between the albedo simulated with the ART theory and observations over smooth surfaces (Dumont et al., 2017; Wang et al., 2017), but real roughness features have always be neglected.

New sections have been added to clearly evaluate RSRT simulations in presence of surface roughness, and to explain why albedo values may be above 1 in the visible range. Overall, a strong effort has been made to make the text clearer (see the track change version of the manuscript). As suggested by the reviewer, we changed the structure of the manuscript by adding two new sections (new 2.4 and 4.1) to explain results with more details.

New Section 4.1:
An entire section has been introduced at the beginning of the Results section to investigate the performance of the RSRT model, before the sensitivity analysis. We think that the accuracy of simulations are more transparent now. In particular, we explain why the simulated spectra do not overlap perfectly observed spectra.

- In the visible, measurements and simulations differ in the 600-700nm range probably because of the presence of impurities which are not considered in simulations (since not measured). Note that the RSRT model is capable of accounting for impurities if they are measured.

- In the NIR domain it is probably because of a small bias in SSA measurements (10% uncertainty). The albedo-SSA relationship in the NIR is linear, meaning that a change of 10% of SSA induces a change of +- 0.01 of albedo. Here, the difference between simulated and measured spectra in the NIR domain is below 0.01 and may come from SSA uncertainties. The impact of these measurement errors in our sensitivity analysis is discussed in detail in Section 4.3.

Hence, measured and simulated spectra differ slightly due to inherent measurement errors. Nevertheless, the RSRT model improves the spectral albedo simulations by taking surface roughness into account, compared to simulations which neglect them (smooth surface, see Figures 6 and 7). Considering all observations, albedo simulations with the RSRT model are improved by a factor 2 by accounting for surface roughness compared to those neglecting them, which is significant.

To the best of our knowledge, this is the first model capable of simulating spectral albedo taking into account the actual surface roughness, the slope and snow optical properties using a Monte Carlo photon transport algorithm.

New Section 2.1 :
We detailed why the measured and simulated albedo values may exceed 1 in the visible range with a new Section in the Methodology section. Explanations are given further in the Major Comments section (#1).

**Major comments**

**#1.** In Fig. 5, all the simulated spectral albedos exceed 1.0 in the wavelength region of < 700 nm even in the case of the flat smooth surface. Also, the measured spectral albedos exceeded 1.0 in the range of < 870 nm in Fig. 7. These results are not realistic in nature and misleading information. Reviewer recommends explaining the reason why spectral albedos exceed 1.0.

The reason why we have albedo values over 1 is because here we consider the apparent albedo, and not the intrinsic albedo (i.e. albedo of a flat terrain): When the terrain is not flat, the horizontal sensor acquiring the snow reflectance is not perfectly parallel to the snow surface, and thus the ratio of the readings from the sensor when measuring the incoming irradiance and the snow reflectance (called the apparent or measured albedo) is different from the intrinsic surface albedo (true albedo = with a perfectly flat surface). Picard et al. (2020) fully detailed these slope effects on the measured albedo, which may be over 1 when the slope is facing the sun, even with perfect instruments and even for weak slopes ~2°. This slope effect, inducing measured spectral albedo in the visible range above 1, has been observed in numerous previous studies (Grenfell et al, 1994 ; Wutttke et al., 2006 ; Dumont et al., 2017), and it has been demonstrated that it is because there is a higher interception probability of the sun beam by these slopes facing the sun compared to the horizontal sensor (Picard et al., 2020). In the present study, the experiments A, B and C have small sun-facing slopes, and it explains why figures show albedo values above 1 in the visible.

Of course, the apparent albedo does not represent the well-known reflectance, the energy is not conserved and it must not be used for energy budget calculations (where a flat terrain has to be considered). It is correct to simulate the apparent albedo here since the goal was to validate RSRT with apparent albedo observations.

To show an example of the slope effect on the measured albedo, Figure 1 illustrates a comparison of two measured spectra acquired over two smooth surfaces in the French Alps in clear sky condition: one with a slope facing away the sun and one with a sun-facing slope. Slopes are below 6°. Snow conditions at the surface were similar, with close SSA values. The presence of the slope facing the sun induces a distortion of the spectra, with values above 1 in the visible and a concave spectral shape. When a small slope is facing back the sun, the pattern of the spectra shows no distortions, and the presence of a slope is difficult to detect. If they are not taken into account, slopes may induce strong biases in snow parameters estimated from optical measurements.

[Figure]

**Figure 1. Measured spectral albedo with Solalb over two smooth titled surfaces having similar snow properties. Measurements are acquired in clear sky conditions. One surface has a 5° slope facing the sun (blue line) and the other has a 3° slope facing away thesun (red line).**

To clearly explain the difference between the apparent and the intrinsic albedo, and the fact that we simulate/measure the apparent albedo in the present study, several changes have been made in the text, and we detail them below.

- The notion of observed apparent albedo is introduced in Section 2.2
L. 180 : « *The observed apparent albedo, hereinafter referred to as $\alpha_{obs}$, is the processed spectrum measured with Solalb, considering the sensor in a horizontal position (Sicart et al., 2001)."*

- We added an entire new section further (now Section 2.4) to explain more clearly the impact of the slope on the observed apparent albedo. The following text has been introduced to illustrate our arguments:
Line 212: « *In the case of a tilted surface, Solalb is not perfectly parallel to the snow surface, and therefore the ratio of values acquired by the sensor when it measures the downwelling and the upwelling spectral irradiance (called the apparent or measured albedo, here αobs) differs from the intrinsic surface albedo (called true albedo in previous studies, Picard et al., 2020). Indeed, when 215 the sensor is horizontal, the titled surface receives sun radiation with a different incidence angle and is viewed with a reduced solid angle by the sensor (Grenfell et al, 1994; Wutttke et al., 2006; Dumont et al., 2017). With surfaces having a sun-facing slope, it has been demonstrated that measured albedo values may be over 1 in the visible range, because there is a higher interception probability of the sun beam by these slopes facing the sun compared to the horizontal sensor (Picard et al., 2020). Therefore, apparent albedo may exceed 1 in the visible range. In contrast, the intrinsic albedo is strictly ranged between 0 and 220 1. In this study, surfaces of experiments A, B and C have small sun-facing slopes (Table 3), and the slope effects can not be neglected in albedo simulations since even a small slope (2°) facing the sun may induce a variation in measured albedo by up to 5 % over a smooth surface (Dumont et al., 2017). »*

- Concerning field experiments, there were no perfect flat surfaces and even the studied smooth surfaces had small slopes (it is very difficult to find a perfectly flat surface in the field). In experiments A, B, C (Fig. 5 and 7) measurements are above 1 in the visible range because surfaces have a sun-facing slope.
L.149 : « *In the reality, it is difficult to find perfectly flat surfaces, and all studied surfaces have small slopes. In particular, it is noteworthly that experiments A, B and C have a small sun-facing slope.*»

- Concerning simulations, we modeled the apparent albedo in order to be compared with measurements. The distorsion of the spectra due to the presence of the slope is modeled with the K factor introduced by Dumont et al. 2017.
L. 270 « *As shown by Dumont et al. (2017), the K factor is the relative change in the cosine of the sun effective incident angle to the slope, and makes it possible to reproduce the distortion of the spectra due to the presence of the slope (with potential albedo values above 1 in the case of a sun-facing slope; Picard et al., 2020)."*

- In the Results section 4.1, we added the following sentence to explain why the albedo values are above 1 when lambda < 700nm :
For Experiments A and B, in Section 4.1, L. 416: «*Both surfaces have a sun-facing slope (3.1° for experiment A and 3.6° for experiment B, see Table 1), so albedo values above 1 in the visible range are not surprising as explained in Sect. 2.4. »*
For experiment C en Section 4.1 : L.447: «*For experiment C, the apparent albedo exceeds 1 in the visible range because of the presence of a sun-facing slope (3.3° - 4°, see Table 1). »*

**#2.** Simulated spectral albedos were not consistent with measured ones as a whole. There are some discrepancies between them. For example, the measured variation $Delta alpha$ shows a clear dependence on $Delta phi\_r$ while the simulated one doesn't (Fig. 8). Reviewer supposes that the measurement values presented here are true. Thus, I am wondering whether the RSRT model provides certain values or not. Authors need to show the agreement between the model and the measurement to present how the proposed model works properly. Otherwise, it could be difficult to achieve the objective of this study which is to quantify the impact of surface roughness on snow albedo.

- Agreements between the model and the measurements are shown in the new section 4.1 'RSRT evaluation'. We show that simulations accounting for the surface roughness are more accurate by about a factor 2 at 700 nm and 1000 nm compared to those neglecting them (i.e ART theory with a flat terrain), which is significant.

L.457: « *Considering all observations, albedo simulations with the RSRT model are improved by a factor 2 by accounting for surface roughness ($\alpha_{sim,rough}$) at 700 nm and 1000 nm compared to those neglecting them ($\alpha_{sim,smooth}$), with an average RMSD of 0.03 at 700 nm and 0.04 at 1000 nm (Table 3). To the best of our knowledge, this is the first model capable of simulating spectral albedo taking into account the actual surface roughness, the topography and snow optical properties using a Monte Carlo photon transport algorithm.*"

In addition, we modified Figures 6 and 7 to illustrate the spectral performance of the RSRT model for all experiments. The differences between measured and simulated albedo spectra are explained as follows :

L. 423: « *For both experiments, the pattern of the measured spectra between 600 nm and 700 nm are probably led by the presence of impurities (not visible to the naked eye in the field). Previous studies showed that even a small concentration of snow LAPs induces a drastic decrease of the albedo in the visible range (Warren, 1984; Dumont et al., 2017), 425 and may explain why measurements and simulations differ in the 600-700nm range. Moreover, the spectra do not overlap perfectly in the NIR domain, but differences are below 0.01, and it is probably because of a small bias in SSA measurements (10% uncertainty). Overall, taking into account the measurement errors, αsim,rough spectra reproduces the observed spectra well for both experiments and the RSRT model improves the spectral albedo simulations by accounting for roughness, compared to those which neglect them (Fig. 5)*"

- To answer to the reviewer about the Fig. 8: there is a dependence between measured variation $Delta alpha$ and $Delta phi\_r$, but it is misleading due to the presence of others contributions that change the albedo and disturb the expect relationship between albedo and surface roughness. It looks like measured $Delta alpha$ increases when $Delta phi\_r$ goes from 42° to 72° (becomes closer to 90°). It means that the roughness effect on albedo values is lower when the roughness orientation is closer to be perpendicular to the sun than parallel. This measured trend is opposite of what we find in the literature, as shown by Warren (1998). The Figure 2 (from Warren, 1998) illustrates the reduction in albedo when the roughness orientation becomes closer to 90°. RSRT simulations reproduce well this trend in Figure 8c and 8d. Authors guess that the measured trend is due to the melting observed in the field, resulting in a smoothing of roughness features during the day. Hence, we can not conclude on this observed trend since several contributions drove the measured albedo. We detailed it in the text as follows :

L. 551: « *In Fig. 8c and 8f, the Δαobs increases when Δφr goes from 42° to 72°, while in theory it should decrease when Δφr approaches 90°. A possible explanation is that melting was observed at the surface in the field, resulting in a smoothing of our roughness shapes during the day, which attenuates the roughness effect on albedo values. Therefore, we cannot conclude on this observed trend since several contributions drove the measured albedo. »*

[Figure]

**Figure 2. Figure 13 in Warren et al (1998). Effect of sastrugi on albedo, from the Monte-Carlo radiative transfer modeling of O'Rawe [1991]. Plotted is the change in albedo as a function of the height-to-width ratio of rectangular sastrugi with spacing equal to width. Simulations are performed with an illumination by a direct beam at 60° from the nadir, for four different sastrugi azimuths; flat-surface albedo = 0.8.**

**General comments**

**1. L29:** Regarding the sentence "For a typical alpine snowpack ... 27 Wm^2).", this estimation was the value at the site C based on the artificial rough surface. Reviewer is wondering if "a typical alpine snowpack" means the natural rough surface in the mountain regions. How does the artificial rough surface represent the natural snow surface in the mountain regions?

The size of the artificial roughness features used in this study are not exaggerated compared to what can be observed in the French Alps. Nevertheless, natural patterns of surface roughness are difficult to quantify in the field since it has high spatial variability in mountainous areas, strongly correlated to the wind, sun exposition, and topography amongst others. Further studies (including *in situ* measurements) are needed to determine what is representative for a natural snow surface in Alpine areas, using photogrammetric measurements for instance, but this is out of the scope of the present study.
To be clearer in the abstract we changed the sentence as follows :
Line 28 : «*For a snowpack where we artificially created surface roughness, we showed that a broadband albedo decrease of 0.05 may cause an increase of the net short wave radiation of 80 % (from 15 W m$^{-2}$ to 27 W m$^{-2}$).* »

**2. L40:** Snow grain shape is also one of the important factor to control the snow albedo (Tanikawa et al., 2006; Jin et al., 2008) in addition to the physical properties mentioned in the manuscript. Authors should add explanations and cite research papers.
- Jin et al. (2008): Snow optical properties for different particle shapes with application to snow grain size retrieval and MODIS/CERES radiance comparison over Antarctica, Remote Sensing of Environment, 112, 3563-3581.
- Tanikawa et al. (2006): Monte Carlo simulations of spectral albedo for artificial snowpacks composed of spherical and nonspherical particles, Applied Optics, 45, 5310-5319.

It is true that the impact of snow grain shape on snow albedo can be significant, and this question has been addressed by the team through several publications (Picard et al., 2009 ; Libois et al., 2013, 2014). We added this factor line 41, as follows (Line 40) :
« *Snow spectral albedo generally depends in a complex way on several factors, including 1) the snow physical and chemical properties, mainly the Specific Surface Area of snow grains (SSA, Gallet et al., 2009), **the snow grain shapes (Tanikawa et al., 2006; Jin et al., 2008; Libois et al., 2013, 2014)** and the concentration of snow Light Absorbing Particles (referred to as LAP, Skiles et al., 2018)* »

And we added the research papers in the reference section.

**3. L199:** What does LAP stand for?

The LAP acronym was described in the introduction Line 42:
« t*he concentration of snow Light Absorbing Particles (called LAP, Skiles et al., 2018)*»

LAP describes several types of impurities such as mineral dust, black carbon or algae. To be clearer, we changed the sentence Line 197 as follows :
« *the concentration of Light Absorbing Particles (called LAP), **such as mineral dust and black carbon,** was not measured although they strongly lower the spectral signature in the visible range (Warren, 1982)* »

**4. L201**: Reviewer is wondering if measured spectral albedo is relatively high at wavelength range 500–700 nm even in a contaminated snow. This comment might be related to the major one.

The high measured albedo in the visible is due to the presence of a small slope facing the sun. This point is explained in detail in the point #1 of the Major Comments section.

**5. L202:** It would be difficult to say a following sentence "The albedo decrease in the 400-600 nm range is a clear structure of a high LAP concentration". Only small amount of black carbon causes a drastic albedo decrease in the visible regions. Authors should add/modify the explanation properly.

It is true that the presence of each LAP type (black carbon, mineral dust, etc.) causes a drastic albedo decrease. To be clearer, we modified the sentence line 201 as follows :
«*The albedo decrease in the 400-600nm range is a clear signature of the presence of snow impurities. Even a small amount of LAP led to a high decrease of the albedo in the visible domain (Tuzet et al 2019)..*»

**6. L205**: Describe the reason why authors chose 700 nm and 1000 nm for the statistical results. The reason is not clear. For example, it would be better to select wavelengths used for satellite remote sensing.

295 Future work will address the application of the model over large-scale natural surfaces and will include atmosphere scattering and absorption, but for now the validity of this model for satellite remote sensing applications is out of the scope of this paper.

The selection of the two wavelengths (700nm and 1000nm) is explained by adding the following sentences:

300 Line 370: "*The main goal of this study is to quantify the roughness effect on albedo values and to determine if this effect is wavelength dependent. Therefore, statistical results are given at two wavelengths: one in the visible domain at 700 nm and one in the NIR domain at 1000 nm. The relation between roughness effect and SSA is investigated at 1000 nm since at this wavelength the albedo sensitivity to SSA is larger (Domine et al., 2006).*"

305 700 nm was chosen randomly but values are relatively stable in the 600-700 nm range.

**7. L210:** How did authors consider the effect of atmosphere in the radiative transfer calculation?

This is explained in the 'simulation framework' section (Sect 3.3). The effects of the atmosphere (scattering and absorption)
310 are not taken into account in the Monte Carlo algorithm. The only atmospheric parameter used in the model is the diffuse-to-total illumination ratio (which depends on atmospheric conditions) to compute the apparent albedo by combining the direct and diffuse albedo components. This parameter was measured in the field shortly after the albedo measurement by screening the sun to record the diffuse irradiance, the total irradiance being measured with the sensor looking upward (see Section 2.2, Line 179).

315

We added the following sentence after the equation 5,
Line 264: « *where $r_{diff-tot}(\lambda, \theta_s)$ is the ratio of diffuse-to-total illumination at wavelength $\lambda$ and at $\theta_s$, measured in the field shortly after each albedo measurements.* »

320 In order to be clearer in the 'simulation framework' section, we modified the sentence :
Line 352: *«RSRT outputs the snow spectral albedo, either in direct or diffuse illumination conditions: $\alpha_{dir,rough}(\lambda, \theta_s)$ and $\alpha_{diff,rough}(\lambda)$, respectively, considering that the plane of the mesh is perfectly flat. Then, $\alpha_{dir,rough}(\lambda, \theta_s)$ and $\alpha_{diff,rough}(\lambda)$ are combined with Eqs. (6) and (7) to simulate the apparent snow albedo of a titled rough surface, called $\alpha_{sim,rough}(\lambda, \theta_s)$, and therefore the simulated apparent albedo accounts for the slope characteristics and surface roughness. Each simulation*
325 *assumes clear sky conditions, 11 and no atmosphere scattering and absorption is considered in the Monte Carlo algorithm. The only atmospheric parameter used in the model is the diffuse-to-total illumination ratio (which depends on atmospheric conditions). This parameter was measured in the field at each albedo acquisition (see Sect.2.1). At the small scale of this study, the effect of the atmosphere is negligible between the sensor and the surface. Future work should add the atmosphere in RSRT for applications over large-scale natural surfaces (mountainous areas)."*

330

**8. L230:** In general, the asymmetry factor (g) increased with increasing (decreasing) the snow grain size (SSA) in the near infrared regions. So, g should be linked with the snow grain size (or SSA). This assumption might lead to biases of spectral albedo simulation.

335 The asymmetry factor (g) is directly linked to the ice particle shapes and degrees of microscopic scale roughness, but g is not measurable in the field or in a laboratory. In the RSRT model we use B (the absorption enhancement parameter) and g to describe the snow grain shape and these parameters are assumed to be constant (i.e. a single homogeneous layer). Nevertheless, we used values adapted for an Alpine snowpack and estimated by Libois et al 2014 as follows :
By combining simulations and measurements of reflectance and irradiance (and not visual observation of snow grains) on an
340 extensive set of snow samples taken in the laboratory and in the field (French Alps and Antarctica), they experimentally found a B value adapted to describe an 'optical grain size'. Then using the correlation between B and 1-g (see Fig. 1 Libois et al. 2014), they deduced g. Thus, they have shown that using B=1.6 and g=0.86 to model snow optical properties is more realistic rather than considering spherical grains as often done.

345 To be clearer, we added the following sentence Line 256:
*«B and g are the snow shape coefficients and are assumed to be constant. Theoretically, g should be directly linked with the wavelength and the ice particle shapes, but as g is not measurable, we used constant values estimated by Libois et al. (2014), who combined simulations and in situ measurements of reflectance in Antarctica and the French Alps. They found that using B = 1.6 and g = 0.86 is more realistic to model snow optical properties rather than considering spherical grains»*
350

Picard et al. 2009 have shown that the uncertainty on SSA measured with reflectance is about 20 % if the snow grain shape is not known. But this value was over-estimated since calculated using two extreme theoretical shapes (spheres/cubics) that are not found in natural snow (which is more like a mixture). In the present study, we assumed that the error on measured SSA to be about 10% (as estimated by Arnaud et al. 2011), and the analysis of the impact of SSA uncertainties on our roughness effect is discussed in Section 4.3.1.

**9. L264:** It is not clear whether the roughness part (Monte Carlo algorithm) employs the single scattering properties (single scattering albedo, phase function and so on) and/or surface reflectance of snow or not. How does the photon decide "hit" or "not hit"? Random number with snow single scattering albedo or snow reflectance? How does next direction after the scattering (i.e. after the photon hits to the snow grain) decide? Detailed explanations are needed.

- To decide if the photon is absorbed or reflected, two configurations are available (KZ04 and Lambertian). The KZ04 configuration employs the single scattering properties while the Lambertian configuration uses a constant surface reflectance of snow (i.e. an ideal diffusion : albedo is the same, whatever the incidence angle). We detailed it in Step 2 (line 310), but to be clearer we introduced this notion earlier by adding the following sentence :
 Line 287: «*Photons are either absorbed or reflected at each hit according to the facet albedo value (Iwabuchi, 2006), that is estimated with the single scattering properties in case of the KZ04 configuration, or as a constant snow reflectance in case of the Lambertian configuration.* »

- How does the photon decide "hit" or "not hit"?
In step 1 we detailed the process of 'hit' or 'not hit' by adding the following sentence :
Line 295: «*Basically, it uses a simple recursive intersection routine to test if the photon hits or does not hit the bounding volume, and when positive, the hitting point is searched using a BVH algorithm (Wald et al., 2007).* »
We added the following the reference:
I*ngo Wald, Solomon Boulos, and Peter Shirley. 2007. Ray tracing deformable scenes using dynamic bounding volume hierarchies. ACM Trans. Graph. 26, 1 (January 2007), 6–es. DOI:https://doi.org/10.1145/1189762.1206075*

- How does next direction after the scattering (i.e. after the photon hits to the snow grain) decide?
Each facet is treated as a snow surface, and the next direction is computed according to the BRDF distribution, depending of the incident angle and snow properties, so the next direction is sensitive to the asymmetry of the scattering. The scattering within a few degrees of the forward direction is much more probable than scattering to other angles (Warren, 1982).
The BRDF computation is detailed in Step 3, and we added explanations with the following sentence :
Line 309: «*Thus, the next direction after the scattering depends 310 of the incident angle of the photon and snow properties. With the KZ04 approximation, the surface is more forward scattering than for a Lambertian surface (Warren, 1982).* »

**10. L463:** In Figs. 8a and d, the results $\Delta alpha$ were not symmetry at $\Delta-phr_r$=0. The effect of surface slope caused the asymmetry of $\Delta alpha$ at $\Delta-phr_r$=0? Explanations are needed.

The effect of surface slope in our sensitivity analysis is reduced by taking a smooth surface with a similar slope to that of the rough surface, so by computing rough-smooth albedo we canceled slope effects. In this case (experiment C), the asymmetry is more an albedo insensitivity to small variations of roughness orientation and it is explained by high SSA values (i.e. lower absorptions). The SSA impact is fully detailed in Section 4.3.1.
Line 534: « *However, for the C rough 90° experiment (Fig. 8b and 8e), Δφr varies from 50° to 122° and Δαobs does not show a strongest albedo reduction around 90°. Similarly, for C rough 0° (Fig. 8a, and 8d), Δαobs values were not symmetrical to Δφr = 0°. This is caused by others contributions that are added to the roughness effects. First, the effect of the slope on albedo varies over time with the solar angle changes. Here we selected a smooth surface with a similar slope to that of the rough surface, so as to minimize the contribution of the slope by comparing rough-smooth albedo at similar illumination conditions (Δαobs). The slope sensitivity to roughness effects is studied in Section 4.3.2. Second, the particularly high values of SSA for this experiment (~100 m2 kg-1 ) induces lower absorption (Warren et al., 1998), and it may explain the albedo insensitivity to small variations of roughness orientation. Moreover, instead of a clear dependence between Δαobs and Δφr, Δαobs pattern shows oscillations, probably caused by the small differences in snow properties between the smooth and the rough surfaces. Indeed, SSA values over the smooth surface are homogeneous, while SSA values over the rough surface evolve unevenly according to the illumination received in the concavities during the day. The SSA sensitivity to roughness effect on albedo measurements is investigated in Section 4.3.1 […]* "*

**11. L635:** This is a rough estimation in a net SW radiation because the validation of the proposed model would not be adequately tested in the visible and shortwave near-infrared region (> 1000 nm). In addition, the effect of snow impurity such as a black carbon and a dust was not considered in the estimation of the net SW radiation. As authors well know, the spectral snow albedo depends on the concentration of snow impurity in the visible region where solar radiation is larger in the relatively cloud free condition. Thus, there would be a large uncertainty in the estimation (there are many parameters to be considered in the estimation, e.g. snow layer (vertical) information). Reviewer supposes that this item is next step.

This is a discussion of the potential albedo impact on the radiative balance. Authors assume that this is a rough estimation in a net SW radiation, with several assumptions, but there is a strong interest to have an order of magnitude of the roughness effect on the absorbed energy.

To be clearer, we modified some sentences, and added some explanations :

Line 29 in the abstract : « *For a snowpack where we artificially created surface roughness, we showed that a broadband albedo decrease of 0.05* **may** *cause an increase of the net short wave radiation of 80 % (from 15 W m$^{-2}$ to 27 W m$^{-2}$).* »

Line 239: « *A simple approach is applied to illustrate the impact of roughness on the quantity of energy absorbed in the snowpack (Sect. 3.4)* »

Line 737: «*The broadband albedo simulated by considering surface roughness is 0.05 lower than the one simulated with the smooth surface. It results to an increase of the SW$_{net}$ from 15 W m$^{-2}$ to 27 W m$^{-2}$ caused by the presence of surface roughness. In other words, the energy absorbed by the snowpack may increase by almost a factor two (+80 %) with the presence of roughness.* **Note that this is an illustration of the potential impact of roughness on the SW$_{net}$, more than a real estimate, because RSRT has not been fully validated at wavelength below 600 nm and above 1050 nm, and because we simulate artificial roughness which may not be representative of the whole alpine snowpack. Nevertheless, these results illustrate the necessity to consider surface roughness in the estimation of the surface energy budget. Further work and measurements are needed to validate the radiative balance simulation, and this is out of the scope of this study***.*»
* * *
Larue and colleagues present both in-situ observations and a Rough Surface Ray Tracer (RSRT) model to assess the impact quantify the impact of surface roughness on snow albedo. Their observations show that surface roughness features have a strong impact at albedo reductions. This impact is already apparent for low roughness values, but becomes more pronounced for higher roughness values, where the albedo reduction depends strongly on the roughness orientation relative to the sun. Besides the observations, Larue and colleagues also introduce for the first time a model that allows to account for surface roughness in snow albedo simulations. Simulations with the model show that albedo simulations are improved by a factor 2 compared to those assuming a smooth surface. The model gives moreover insight in the role of Specific Surface Area (SSA), slope, the solar zenith angle and the roughness orientation. Finally, the paper highlights the necessity to take into account the roughness effects to compute the surface energy budget.

**GENERAL COMMENTS**

The paper of Larue and colleagues touches upon an important topic, is well written, extensively analyzed. As such it build further on earlier work of Warren, Cathles, Pfeffer, Lhermitte and many others, but with the clear novelty that it adds new well designed measurements and the RSRT model that allows to assess the effects in 3D (versus earlier 2D models). Based on these comments I think the paper is well suited and already well written and organised to merit publication in TC. Nevertheless, I have some minor comments that might be addressed in an eventual revised version of the paper.

**MINOR COMMENTS**

**L124** "by uniformly pressing a rectangular metal bar into the snow" : What would be the effect of compression and corresponding differences in density/SSA on the observed albedo values. Do you expect this to interfere with the observations? If so/not, why and what would be the effect?

By compacting the snow, we locally increase the snow density at the surface and it may lead to a small decrease of the SSA (Legagneux et al., 2002 ; Domine et al., 2007..). It is true that for the experiments A and B, we may have disturbed surface SSA observations in the concavities by pressing the bar into the snow. But as the compaction was weak (2cm), and the SSA values were small (7.2 and 4.5 m²/kg before the compaction), we can consider that the observed albedo values were not, or weakly, affected by the compaction.
To be clearer, we added the following sentence line 190:

*« Note that compacting to create the roughness features may have lowered the SSA locally. As the compaction was small (2 cm depth), and as the SSA values were initially low over the studied surfaces, we assumed here that the effect of the compaction on the observed albedo is negligible. "*

For experiment C and D, we measured 3 surface SSAs at each albedo acquisition to have a good representativeness, with samples taken 1) over the side of the cavity facing the sun, 2) over the side of the cavity facing away the sun, and 3) over the smooth surface between the cavities. The differences of SSA measured over each surface are lower than 10 %. We took the mean SSA to compute the albedo, as for each studied area the percentage of surfaces facing, facing away from the sun and smooth are similar. To be clearer on this point we added the following sentence in the paper (line 192) :
*"For the two experiments C and D, three SSA measurements were taken at the surface at each albedo acquisition: two in the cavities (one over the side facing the sun, one over the side facing away the sun) and one over the smooth surface between cavities. The standard deviations of these three SSA are always lower than 10% of the mean SSA, showing that the compaction effect is negligible compared to measurements uncertainties. The mean of these three SSA values is used in our albedo simulations."*

**Measured albedo values above 1:** the paper shows several figures with spectral albedo values above 1 which is physically impossible. It would be good to explain where these values come from and what it means in terms of uncertainty (also for the rest of the observations and conclusions).

See comment #1 of Reviewer 1. We introduced a new section (now Section 2.4) to fully explain why albedo values may exceed 1 in the visible range. It is because we measured the apparent albedo (with a sensor placed horizontally, over a tilted terrain), and this is different from the true albedo (strictly ranges between 0-1, with a perfectly flat surface). In the present study, the presence of small slopes facing the sun leads to apparent albedo values above 1 in the visible.

**Figure 1:** Based on this figure it seems that the sun is oriented North. I know that it
is only an illustration and a minor detail, but it might be clearer if the sun is positioned south for norther hemisphere experiments.

We modified the Figure 1 to position the sun South in the illustration (see new Fig. 1).

**Figure 5:** Comparison between the simulated smooth and observed albedo values seems to show still some minor contamination by LAP's in shorter wavelengths. Perhaps worthwhile to mention that as well when discussing this graph?
This is true, we changed the sentence line 423 to mention the weak contamination by LAP's in shorter wavelengths :
*"For both experiments, the pattern of the measured spectra between 600 nm and 700 nm are probably led by the presence of impurities (not visible to the naked eye in the field). Previous studies showed that even a small concentration of snow LAPs induces a drastic decrease of the albedo in the visible range (Warren, 1984; Dumont et al., 2017), and may explain why measurements and simulations differ in the 600-700nm range. "*

**L650** "large scale": it would be good if the authors could already add a discussion point of what the current results would mean for larger scale roughness features and/or how the conclusions from this paper can (or not) be extrapolated to larger scale roughness features.

The RSRT model can be used at a larger scale if it is driven by an adapted DEM. Nevertheless, at this large scale, the most challenging work would be to include the atmospheric effects in the Monte Carlo algorithm.
To be clearer on this point, in the simulation framework section we explain that the atmosphere effects are not directly taken into account in the Monte Carlo algorithm, by adding the following sentence Line 355:
*"Each simulation assumes clear sky conditions, 11 and no atmosphere scattering and absorption is considered in the Monte Carlo algorithm. The only atmospheric parameter used in the model is the diffuse-to-total illumination ratio (which depends on atmospheric conditions). This parameter was measured in the field at each albedo acquisition (see Sect.2.1). At the small scale of this study, the effect of the atmosphere is negligible between the sensor and the surface. Future work should add the atmosphere in RSRT for applications over large-scale natural surfaces (mountainous areas)."*

To discuss the future work concerning the model adaptation at larger scale, we added the following sentences at the end of the section 4.5, line 745:

[revised manuscript text omitted]

---

## Author Response (AR2)

**Letter to the editor**

**Revision of manuscript number (doi)**: 10.5194/tc-2019-179-author_response-version2.pdf

Dear Editor-in-chief,

Please find below to this letter the marked-up version of the manuscript. In preparing the final draft for publication, we carefully reviewed the manuscript to cancel the minor grammatical issues. Moreover, we carefully compiled the reference list according to the TC standardds.

We would like to thank the reviewers and the editor for their detailed comments which were useful to complete this study.

Please, do not hesitate to contact me if you have any further questions or comments, or if anything is missing from the submission of the revised manuscript.

Regards,

Fanny Larue
* * *

[revised manuscript text omitted]

Wuttke, S., Seckmeyer, G., and& König-Langlo, G.: (2006). Measurements of spectral snow albedo at Neumayer, Antarctica,. In Annales Geophysicae (Vol. 24), 2006,. Retrieved from http://lap.physics.auth.gr/

Zege, E., P., Ivanov, A., P., and Katsev, I., L.: Image transfer through a scattering medium, Berlin: Springer, 1991,

Zhuravleva, T. B., and Kokhanovsky& Kokhanovskii, A. A.: (2010). Influence of horizontal inhomogeneity on albedo and
  absorptivity of snow cover,. Russian Meteorology and Hydrology, 35(9), 590–595.
  https://doi.org/10.3103/S1068373910090025, 2010.

Zhuravleva, T. B., and& Kokhanovsky, A. A.: (2011). Influence of surface roughness on the reflective properties of snow,.
  Journal of Quantitative Spectroscopy and Radiative Transfer, 112(8), 1353–1368.
  https://doi.org/10.1016/J.JQSRT.2011.01.004, 2011